# Cellular reprogramming in vivo initiated by SOX4 pioneer factor activity

Takeshi Katsuda [1,2,3,8], Jonathan H. Sussman[1,2,3,4], Kenji Ito[1,5], Andrew Katznelson[1,5], Salina Yuan[1,2,3], Naomi Takenaka[1,5], Jinyang Li[1,2,3], Allyson J. Merrell[1,2,3], Hector Cure[1,2,3], Qinglan Li[1,5], Reyaz Ur Rasool[1,3,6,7], Irfan A. Asangani [1,3,6,7], Kenneth S. Zaret [1,5,7] ✉ & Ben Z. Stanger [1,2,3,5] ✉

Tissue damage elicits cell fate switching through a process called metaplasia, but how the starting cell fate is silenced and the new cell fate is activated has not been investigated in animals. In cell culture, pioneer transcription factors mediate "reprogramming" by opening new chromatin sites for expression that can attract transcription factors from the starting cell's enhancers. Here we report that SOX4 is sufficient to initiate hepatobiliary metaplasia in the adult mouse liver, closely mimicking metaplasia initiated by toxic damage to the liver. In lineage-traced cells, we assessed the timing of SOX4-mediated opening of enhancer chromatin versus enhancer decommissioning. Initially, SOX4 directly binds to and closes hepatocyte regulatory sequences via an overlapping motif with HNF4A, a hepatocyte master regulatory transcription factor. Subsequently, SOX4 exerts pioneer factor activity to open biliary regulatory sequences. The results delineate a hierarchy by which gene networks become reprogrammed under physiological conditions, providing deeper insight into the basis for cell fate transitions in animals.

Metaplasia is an adaptive cellular response to tissue injury in a variety of organs, including the lung (squamous metaplasia), esophagus (intestinal metaplasia), and pancreas (acinar-ductal metaplasia)[1,2]. Metaplasia involves the induction of multiple transcription factors[3–6], and loss-of function studies in vivo suggest that these factors function in concert with epigenetic remodeling[3–8]. However, whether the induced transcription factors are sufficient to elicit metaplasia and how they might initiate cell fate changes in vivo is not known. During metaplastic transitions, the genetic network of the starting cell fate is suppressed while the new cell fate is activated by complex mechanisms which have yet to be elucidated.

Pioneer transcription factors possess the ability to elicit cell fate changes by targeting silent genes bound by nucleosomes in regions of repressed chromatin[9]. Conversely, pioneer factors can also enable further chromatin compaction and repression[10] and are involved in diverse processes ranging from cancer progression to the expression of circadian rhythm genes[11,12]. Pioneer factors can trigger reprogramming of one cell type to another in cultured cells – notable examples include OCT4, SOX2, and KLF4 in induced pluripotent stem cells (iPSCs)[13–16], FOXA1/2/3 in induced hepatocytes[17], and ASCL1 and NEUROD1 in induced neurons[18,19]. In cultured cells, repression of the native cell type regulatory genes is thought to occur as the reprogramming factors bind to regulatory elements specific to the target cell type fate. In the process, they disengage transcription factors from active enhancers[13,20] that are responsible for maintaining the initial cell state, a process known as enhancer decommissioning[21,22].

[1]Perelman School of Medicine, University of Pennsylvania, Philadelphia, PA, USA. [2]Department of Cell and Developmental Biology, University of Pennsylvania, Philadelphia, PA, USA. [3]Abramson Family Cancer Research Institute, University of Pennsylvania, Philadelphia, PA, USA. [4]Graduate Group in Genomics and Computational Biology, Perelman School of Medicine, University of Pennsylvania, Philadelphia, PA, USA. [5]The Institute for Regenerative Medicine, University of Pennsylvania, Philadelphia, PA, USA. [6]Department of Cancer Biology, University of Pennsylvania, Philadelphia, PA, USA. [7]Penn Epigenetics Institute, University of Pennsylvania, Philadelphia, PA, USA. [8]Present address: Department of Chemical System Engineering, Graduate School of Engineering, The University of Tokyo, Tokyo, Japan. ✉e-mail: zaret@pennmedicine.upenn.edu; bstanger@upenn.edu

We and others have reported that the liver undergoes a hepato-biliary metaplasia ("biliary reprogramming") as a conserved in vivo response to liver injury, wherein hepatocytes are reprogrammed to become biliary epithelial cells[23–26]. Notably, the liver injury model allows us to trace and isolate individual cells undergoing metaplasia at different time points of the process. Here, we use the hepatobiliary metaplasia model, as well as an in vivo ectopic gene expression system, to reveal the underlying genetic regulatory mechanisms involved in the cell fate change. Our studies provide insight into the coordination of gene activation and repression programs responsible for the reprogramming process in regenerating mammalian tissues.

## Results

### *Sox4* and *Sox9* are induced during biliary reprogramming

We and others have reported that in response to cholestatic injury induced with a 0.1% 3,5-diethoxycarbonyl-1,4-dihydrocollidine (DDC) diet, adult mouse hepatocytes undergo hepatobiliary metaplasia (reprogramming) through a series of stepwise phenotypic changes[24,25,27]. The studies suggested that CD24 would serve as a useful surface marker of cells at the early-to-intermediate stages of repro-gramming and EPCAM as a surface marker of cells at the intermediate-to-late stages of reprogramming[27]. We confirmed the predictions by performing flow cytometry (Fig. 1a, Supplementary Fig. 1) and bulk RNA-Seq (Fig. 1b, c) of liver cells isolated from mice treated with DDC for 4–13 weeks. The results shown validated the utility of CD24 and EPCAM for identifying and isolating cells at different stages of injury-induced hepatobiliary reprogramming (Supplementary Figs. 2 and 3).

Our previous studies indicated that during the reprogramming process, hepatocytes alter their chromatin landscape to resemble that of biliary epithelial cells. Notably, newly opened chromatin regions were highly enriched for SOX binding motifs[27]. Given that SOX tran-scription factors are known to possess pioneer factor activity in cell culture[15,28], we hypothesized that one or more SOX factors facilitates biliary reprogramming in vivo by directly eliciting chromatin accessi-bility. Using the RNA-Seq data, we found that *Sox4* and *Sox9* are the only *Sox* factors to be expressed during injury-induced reprogram-ming (Supplementary Fig. 4a). *Sox9* was weakly expressed in normal hepatocytes (Fig. 1d, Supplementary Fig. 4b), as previously reported in a subpopulation of periportal hepatocytes[29], while *Sox4* expression was virtually undetectable in hepatocytes at baseline but rapidly induced during reprogramming (Fig. 1d, Supplementary Fig. 4b). This corresponded to a ~20-fold increase of *Sox9* expression in repro-grammed cells compared to control hepatocytes, while *Sox4* expres-sion increased by ~3000-fold compared to control hepatocytes.

### *Sox4* and *Sox9* are necessary for hepatobiliary reprogramming

We first set out to explore whether either *Sox4* or *Sox9* are required for biliary reprogramming through loss-of-function experiments in Rosa26-LSL-Cas9-EGFP mice[30,31]. We also examined *Hnf1b* as a control for this system, as it is known to be required for normal bile duct development[32]. Accordingly, we systemically injected into the mice serotype 8 adeno-associated virus (AAV8) packaged with TBG-*Cre* and either gene-targeting sgRNAs or no additional payload (empty vector; EV), which results in hepatocyte-specific activation of Cas9 and EGFP, with consequent gene disruption with targeted sgRNAs. One week after AAV injection, mice were challenged with 0.1% DDC to induce biliary reprogramming (Fig. 1e). The AAV8 system enables robust infection of >95% of hepatocytes and, significantly, allows Cas9 acti-vated cells at different stages of reprogramming to be recognized and isolated by virtue of an EGFP lineage tracer. Flow cytometry performed 9 weeks after the start of DDC treatment confirmed reduction of reprogramming efficiency in all the knockout (KO) animals (Fig. 1f). Interestingly, *Sox9* KO suppressed both early and late stages of reprogramming (as reflected by CD24 and EPCAM positivity, respec-tively), while *Sox4* KO suppressed only the early stage (Fig. 1f).

Strikingly, *Sox4* and *Sox9* double-KO (DKO) almost completely blocked reprogramming at both early and late stages to a degree comparable or greater than *Hnf1b* deletion, suggesting a synergistic effect of *Sox4* and *Sox9* in the reprogramming process (Fig. 1f). This interpretation was confirmed at a global transcriptional level with bulk RNA-Seq of FACS-sorted EGFP+ cells (Fig. 1g). Additionally, gene set enrichment analysis (GSEA) using our previously established hepatocyte-enriched and reprogramming-enriched signatures[27] validated the inhibition of reprogramming in KO cells at the whole transcriptome level (Fig. 1h). Collectively, these results show that the combined activity of *Sox4* and *Sox9* are necessary for hepatobiliary reprogramming.

### *Sox4* induces biliary reprogramming

We then tested whether ectopic expression of *Sox4* and/or *Sox9* – delivered via AAV gene transfer – are sufficient to initiate biliary reprogramming of hepatocytes under toxin-free conditions. To this end, we produced AAV8-TBG-*HA-Sox4*-P2A-*Cre* and AAV8-TBG-*HA-Sox9*-P2A-*Cre* vectors and injected viral preps individually or con-currently to Rosa26-LSL-*Cas9-EGFP* mice (Fig. 2a). As a control, mice were injected with empty vector (AAV8-TBG-*Cre*).

We harvested hepatocytes at 7 days post injection (dpi) and confirmed increased expression of both *Sox4* (~1000–1500-fold) and *Sox9* (~10–30-fold) by qRT-PCR (Fig. 2b). Importantly, both of the fold increases were comparable to those observed in reprogrammed or biliary cells undergoing DDC-induced reprogramming (Supplemen-tary Fig. 4b). Immunofluorescence (IF) confirmed both SOX4 and SOX9 expression at the protein level (Fig. 2c, Supplementary Fig. 5a). SOX4 expression was further confirmed using an anti-HA tag antibody (Supplementary Fig. 5b). Therefore, we concluded that our expression system reasonably recapitulates the relative levels of upregulation of SOX4 and SOX9 observed under physiological conditions.

Strikingly, flow cytometry after 7 days demonstrated that ectopic expression of *Sox4* or *Sox4+Sox9* (abbreviated *Sox4/9*) together, but not *Sox9* alone, induced robust expression of CD24 and modest expression of EPCAM in EGFP+ hepatocyte-derived cells (Fig. 2d, Supplementary Fig. 6). A broader analysis of gene expression by qRT-PCR confirmed the induction of multiple biliary genes and the repression of multiple hepatocyte genes following ectopic expression of *Sox4* and *Sox4/9*, while *Sox9* alone had no effect (Fig. 2e). We also noted differences in the expression kinetics of *Sox4* and *Sox9*. Speci-fically, *Sox4* mRNA expression reached its maximum level from days 1–4 dpi, then began to diminish slightly by 7 dpi, and was greatly decreased by 10 dpi (Fig. 2f). By contrast, *Sox9* mRNA was initially expressed at low levels and increased over a 2 week period (Supple-mentary Fig. 7a). Despite the delayed and prolonged induction of *Sox9* relative to *Sox4*, we found no evidence that ectopic expression of *Sox9* could induce detectable changes in the expression of major hepato-cyte/biliary marker genes (Supplementary Fig. 7b). Therefore, we focused our subsequent attention on *Sox4*.

*Sox4* mice lost weight following viral induction, reaching a nadir at 7-8 dpi before recovering most of the weight by 14 dpi (Supplementary Fig. 7c). During this period, the liver became smaller and paler than empty vector-injected controls (Supplementary Fig. 7d), with atypical, ectopic ductal cells observed (Fig. 2g, arrows), a hallmark of chronic and fulminant liver diseases[33]. Consistent with the CD24 to EPCAM reprogramming sequence identified by flow cytometry in DDC-treated mice (Fig. 1a), we found that CD24 was robustly expressed at 4 dpi in *Sox4* hepatocytes at the protein level, whereas EPCAM became strongly expressed only at 7 dpi (Fig. 2h). Other intermediate-to-late biliary reprogramming markers, such as PROM1 and ITGA6, became detectable between 4 dpi and 7 dpi (Fig. 2h). We did not observe EGFP+ cells that co-expressed KRT19, a marker of fully reprogrammed cells[27], suggesting that the biliary induction by SOX4 is limited to the early stage of biliary reprogramming. Consistent with these data, flow cytometry demonstrated that CD24 expression peaked at 4 dpi,

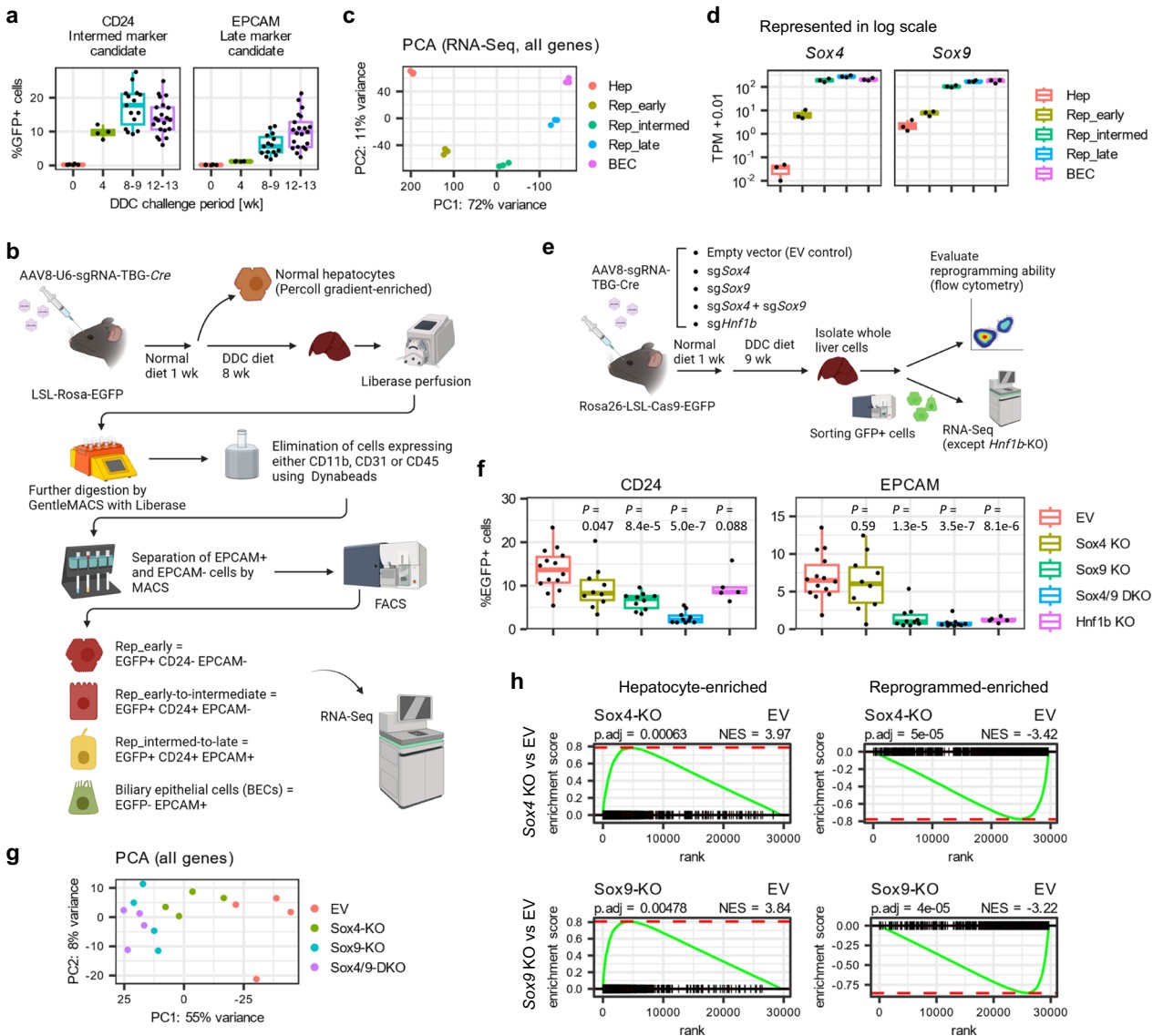

**Fig. 1 | *Sox4* and *Sox9* are required for biliary reprogramming. a** Flow cytometry for CD24 and EPCAM at the indicated time points (weeks) following DDC challenge (*n* = 4, 4, 15, and 21 animals for 0, 4, 8–9, and 12–13 wpi, respectively). **b** Schematic showing the strategy for isolating early reprogrammed cells (EGFP+CD24− EPCAM−), early-to-intermediately reprogrammed cells (EGFP+CD24+EPCAM−), and intermediate-to-late reprogrammed cells (EGFP+CD24+EPCAM+). **c** PCA of whole transcriptomes (RNA-Seq) for the indicated cell populations along the hepatocyte-to-biliary axis (*n* = 3 animals). **d** Results of RNA-Seq showing the expression of *Sox4* and *Sox9* during DDC-induced biliary reprogramming. Expression levels are normalized by TPM (transcripts per million) (*n* = 3 animals). **e** Schematic for the design of the loss-of-function experiment. AAV8 packaged with three sgRNAs targeting *Sox4* and/or *Sox9*, and *Hnf1b* along with TBG-*Cre* was injected to Rosa26-LSL-*Cas9*-*EGFP* mice to induce knockout, while enabling the simultaneous genetic labeling of the infected hepatocytes. Biliary reprogramming was then induced by introducing

a DDC diet. Animals were sacrificed for analysis 9 weeks later. **f** Reprogramming efficiency was assessed by flow cytometry using two surface markers, CD24 and EPCAM, which serve as surrogates for early and late reprogrammed cells, respectively (*n* = 14, 10, 10, 10, and 5 animals for empty vector, *Sox4* KO, *Sox9* KO, *Sox4/9* DKO, and *Hnf1b* KO conditions, respectively). Exact *P* values by two-sided t-test vs. the empty vector (EV) group as reference are shown without adjustment. **g** PCA mapping of the RNA-Seq data of the sorted EGFP+ cells (*n* = 4 animals). **h** GSEA for the RNA-Seq data using hepatocyte-enriched and reprogrammed cell-enriched signatures (*n* = 4 animals). The signature gene sets were curated using previously published RNA-Seq data[27] by comparing the gene expression of normal hepatocytes and DDC-induced reprogrammed cells (YFP+EPCAM+ cells). Adjusted *P* values and NES values were calculated using the R fgsea package. (**a**, **d**, **f**) The center line, box limits, whiskers, and points indicate the median, 25th/75th quartiles and 1.5× interquartile range, respectively.

becoming detectable in over 60% of EGFP+ hepatocytes before returning to baseline, while EPCAM expression in EGFP+ cells continued to increase after 4 dpi (Supplementary Fig. 7e).

We then performed RNA-Seq using *Sox4*-expressing and control hepatocytes harvested at 4 dpi. We first compared *Sox4* expression levels in *Sox4*-expressing hepatocytes to those of DDC-induced reprogrammed and biliary cells, and again confirmed comparable induction, with slightly greater *Sox4* expression observed following AAV infection (Supplementary Fig. 8a). Importantly, PCA projection of

RNA-Seq data demonstrated that *Sox4*-expressing hepatocytes exhibited a markedly similar degree of reprogramming compared to DDC-induced early reprogrammed cells, as revealed by the PC1 axis capturing 68% of the variance (Fig. 2i), and similar inter-replicate versus intra-replicate correlations between *Sox4*-expressing hepatocytes and DDC-induced reprogrammed cells (Supplementary Fig. 8b). Using gene sets built from differentially expressed genes (see Methods) between hepatocytes and Rep_early cells (2355 upregulated and 1189 downregulated genes in Rep_early vs. hepatocytes) (Supplementary

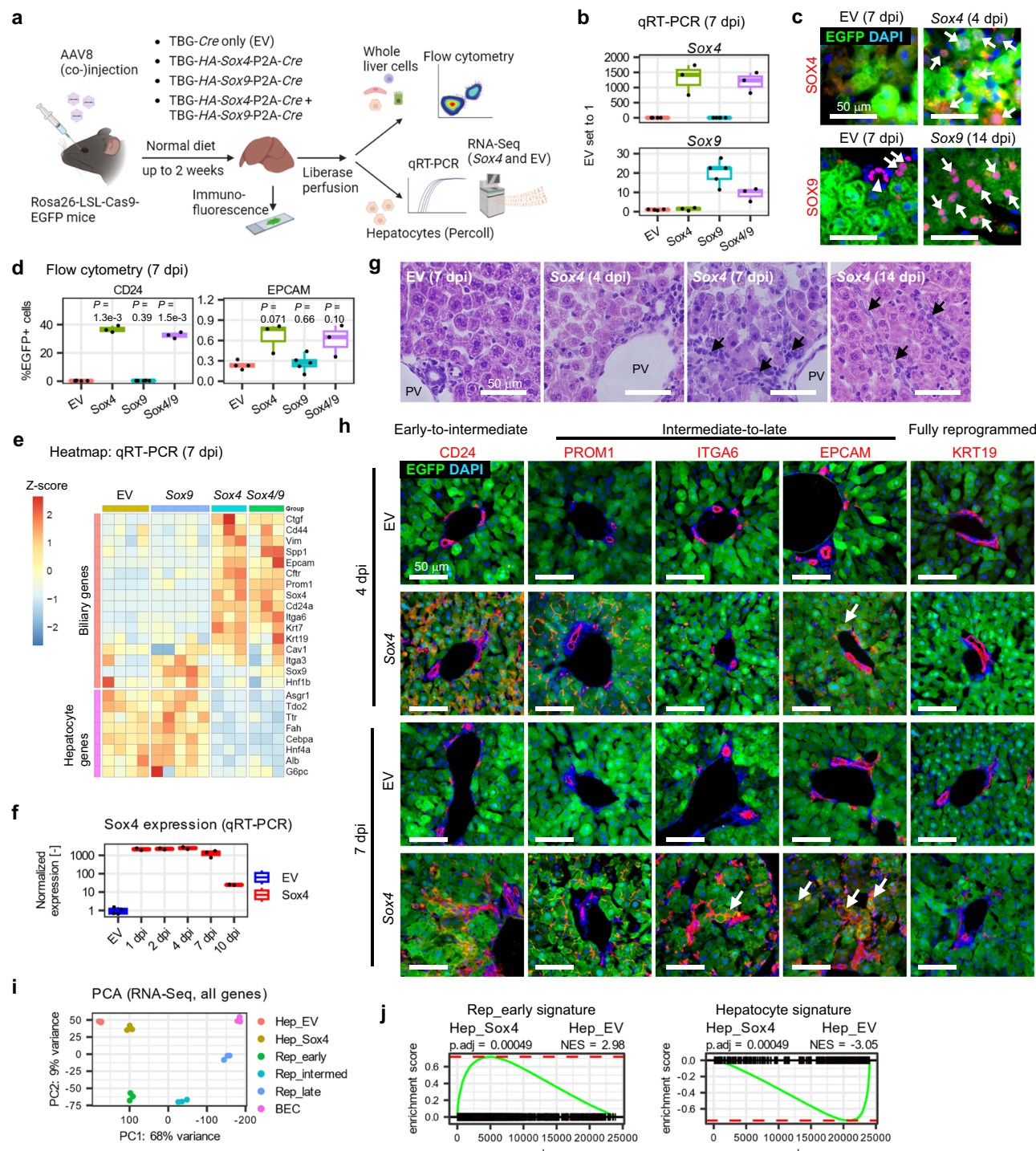

**Fig. 2 | Ectopic expression of *Sox4* is sufficient to induce the early stage of biliary phenotypes in adult hepatocytes. a** Experimental design. AAV8 packaged as shown was injected into Rosa-LSL-*Cas9-EGFP* mice. Total liver cells or enriched hepatocytes were harvested after 2 weeks on normal diet. **b** qRT-PCR of *Sox4* and *Sox9*. **c** IF of SOX4 and SOX9 at the designated time points. Arrows indicate nuclear staining of SOX4 or SOX9 in EGFP+ hepatocyte-derived cells. Arrowhead indicates SOX9 staining in BECs. **d** Reprogramming efficiency assessed by flow cytometry of CD24 and EPCAM. Exact *P* values by two-sided t-test with the empty vector (EV) group as reference are shown without adjustment. **e** Heatmap of qRT-PCR for biliary and hepatocyte genes. Expression levels are normalized to *Actb* and shown as z-scores for each gene. **f** Kinetics of *Sox4* mRNA expression assessed by qRT-PCR (*n* = 2, 2, 2, 3, and 2 animals for *Sox4* samples at 1, 2, 4, 7, and 10 dpi, respectively). Data for the EV control pooled from 4, 7, and 10 dpi (*n* = 4 animals). **g** H&E of AAV-*HA-Sox4*-injected or control livers at designated time points. Arrows indicate

atypical ductal cells extending from portal vein (PV) regions. **h** Representative IF of reprogramming markers in livers harvested from EV- or *Sox4*-injected mice at the designated timepoints. Arrows indicate EGFP+ hepatocyte-derived cells expressing the indicated marker. **i** PCA plot of RNA-Seq data from EV- and *Sox4*-injected mice compared to reprogrammed cells following DDC treatment (*n* = 3 animals). **j** GSEA comparing EV and *Sox4* hepatocytes, using the hepatocyte-enriched and early reprogrammed cell (Rep_early)-enriched signatures (*n* = 3 animals). Adjusted *P* values and NES values were calculated using the R fgsea package. **b**, **f** Expression normalized to *Actb*, with EV hepatocyte expression set to one. **b**, **d**, **e**: *n* = 4, 3, 5, and 3 animals for EV, *Sox4*, *Sox9* and *Sox4/9* conditions, respectively. **b**, **d**, **f** The center line, box limits, whiskers, and points indicate the median, 25th/75th quartiles and 1.5× interquartile range, respectively. **c**, **g**, **h** Scale bar = 50 µm. Staining experiments used samples from at least 2 animals.

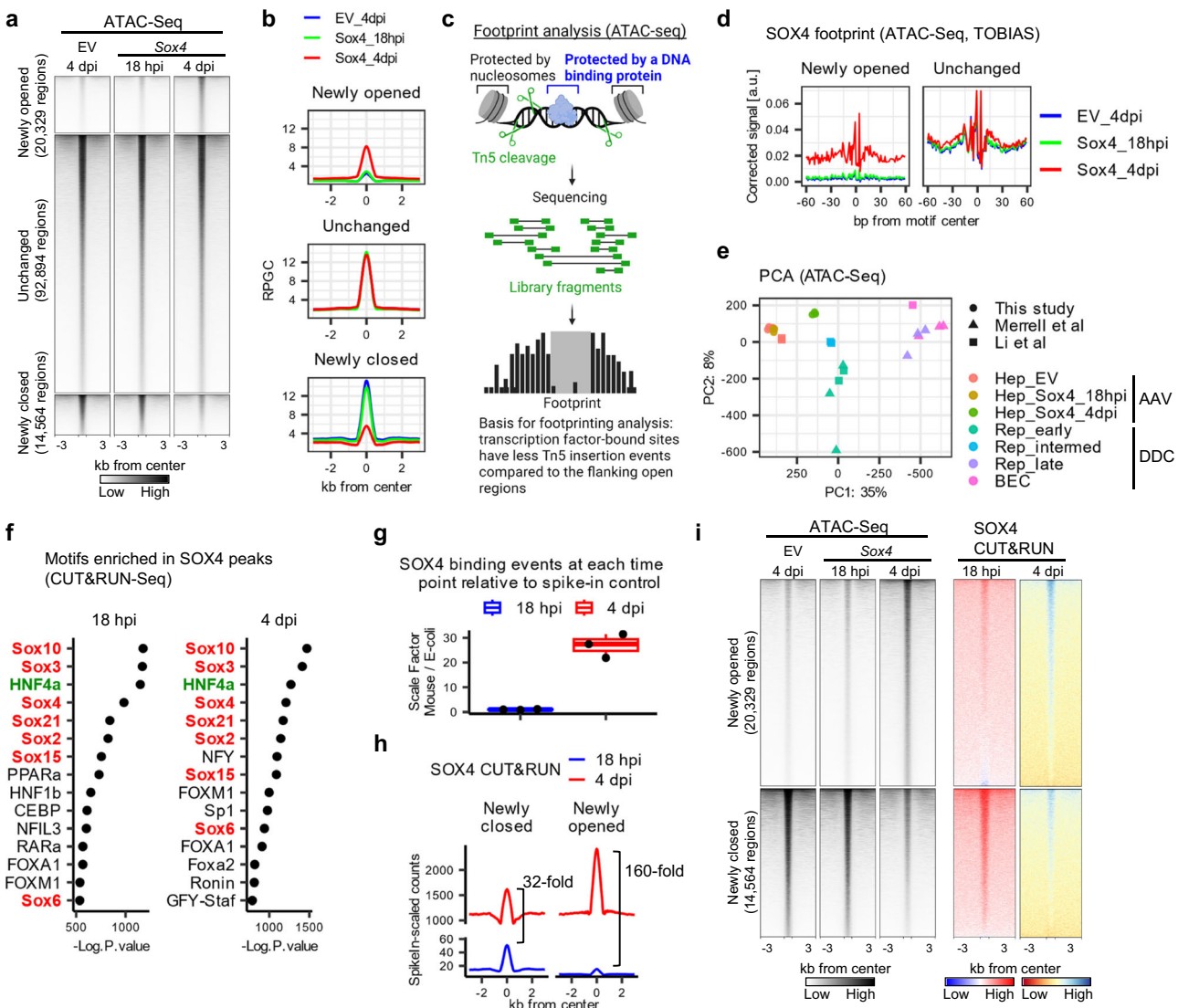

**Fig. 3 | The chromatin landscape following ectopic expression of *Sox4* recapitulates that of partially reprogrammed cells. a** Differential peak analysis at 4 dpi showing newly opened, newly closed, and unchanged regions following *Sox4* expression in hepatocytes compared with empty vector (EV). **b** Averaged aggregate plots of the ATAC-Seq signals in the newly opened, unchanged, and closed regions corresponding to panel **a**. **c** Schematic describing ATAC-Seq-based footprinting as implemented by TOBIAS[35]. **d** Results of DNA footprinting analysis for the SOX4 binding motif comparing empty vector and *Sox4* hepatocytes at newly opened and unchanged regions as defined in (**a**). **e** PCA mapping for ATAC-Seq of empty vector EV and *Sox4* hepatocytes (*n* = 3 animals) at 4 dpi. Data obtained for this study were downsampled by approximately 1/100-fold to adjust read depth and compared with previously published data[27,36]. "Rep_intermed" indicates SOX9+ cells sorted from *Sox9-RFP* reporter mice treated with DDC[36]. **f** Results of HOMER motif analysis using all SOX4 peaks from CUT&RUN-Seq (*n* = 9,463 at 18 hpi; *n* = 19,362 peaks at 4

dpi). The top 15 motifs from each timepoint ranked as *p*-values are shown. SOX motifs, highlighted in red, were highly ranked. The HNF4A motif, highlighted in green, was the third most significantly enriched motif at both 18 hpi and 4 dpi. **g** Quantification of SOX4 total binding was estimated using E. coli spike-in genomic DNA for scaling. Data are shown as fold-change compared to mean values at 18 hpi data (*n* = 3 animals). The center line, box limits, whiskers, and points indicate the median, 25th/75th quartiles and 1.5× interquartile range, respectively. **h** Averaged aggregate plots for spike-in-scaled SOX4 CUT&RUN-Seq data at newly closed and newly opened regions at the designated time. **i** SOX4 CUT&RUN signals (right two columns) visualized as heatmaps for the newly opened and newly closed regions defined by ATAC-Seq (left three columns). Corresponding averaged aggregate plots for SOX4 CUT&RUN are shown in Supplementary Fig. 11b. Note that SOX4 data are normalized genome-wide in each sample (18 hpi and 4 dpi) and do not support quantitative comparison between the two timepoints.

Data 1), GSEA confirmed that *Sox4*-expressing hepatocytes were significantly enriched for a reprogrammed cell signature and de-enriched for the hepatocyte signature (Fig. 2j). Taken together, the data indicate that *Sox4* is sufficient to induce global transcriptional changes that recapitulate the initial stages of biliary reprogramming induced by DDC, including the repression of hepatocyte gene expression.

### SOX4 remodels chromatin landscapes
Based on these observations, we hypothesized that SOX4 elicits the early stages of hepatobiliary reprogramming by altering the chromatin landscape. As such, we performed Assay for Transposase-Accessible

Chromatin using sequencing (ATAC-Seq)[34] with hepatocytes isolated at 18 h post injection (18 hpi) and 4 dpi from AAV-*HA-Sox4*-injected mice as well as AAV-EV-injected mice at 4 dpi as a control. Hepatocytes at 18 hpi showed weak HA-SOX4 expression at the protein level (Supplementary Fig. 9a) and minimal changes in gene expression (Supplementary Fig. 9b), thereby providing a profile of early SOX4 binding. Differential peak analysis comparing *Sox4*-expressing hepatocytes to control hepatocytes at 4 dpi identified 20,329 regions with increased accessibility ("newly opened regions"), 14,564 regions with decreased accessibility ("newly closed regions"), and 92,894 regions exhibiting no change ("unchanged regions") (Fig. 3a, b). By contrast, SOX4

expression had almost no effect on the chromatin profile at 18 hpi (Fig. 3a, b).

Owing to the protective effects of DNA-bound transcription factors against cleavage by Tn5 transposase, analysis of transposase cleavage patterns in deeply sequenced samples enables visualization of transcription factor footprints within ATAC peaks. More specifically, we utilized a recently developed footprinting analysis tool, TOBIAS (Transcription factor Occupancy prediction By Investigation of ATAC-Seq Signal) (Fig. 3c)[35], which enables highly accurate prediction of transcription factor binding. As expected, the SOX4 footprint was enriched in newly opened regions, while no difference was observed in the unchanged regions (Fig. 3d). When compared with previously published ATAC-Seq studies from our group[27] and others[36], *Sox4*-expressing hepatocytes at 4 dpi exhibited open chromatin profiles resembling early reprogrammed cells (Fig. 3e, Supplementary Fig. 9c, Supplementary Fig. 10), consistent with our transcriptome-based findings with RNA-Seq (Fig. 2i, j). Thus, global changes in the chromatin landscape resulting from ectopic expression of *Sox4* are similar to those observed during early stages of DDC-induced biliary reprogramming.

## SOX4 binding precedes major changes in gene expression
To explore the association between SOX4 binding and chromatin opening, we performed a genome-wide examination of SOX4 binding patterns. To this end, we performed Cleavage Under Targets & Release Using Nuclease sequencing (CUT&RUN-Seq)[37,38] using hepatocytes isolated at 18 hpi and 4 dpi (Supplementary Fig. 11a). We obtained 9,463 and 19,362 SOX4 peaks at 18 hpi and 4 dpi samples respectively (see Methods). As predicted, the peaks were highly enriched for SOX binding motifs (Fig. 3f).

Using an E. coli DNA spike-in approach, we quantitated SOX4 binding at each time point. As expected, the absolute amount of SOX4 binding increased from 18 hpi to 4 dpi (Fig. 3g). Surprisingly, this increase in binding was observed both in newly opened and newly closed regions (Fig. 3h). The increase in SOX4 binding from 18 hpi to 4 dpi was greatest in the newly opened regions, where the averaged peak summit (4 dpi/18 hpi) increased by 160-fold; in newly closed regions, by contrast, binding increased by 32-fold (Fig. 3h). The trends were also seen when the data were normalized across the genome (Fig. 3i, Supplementary Fig. 11b). Because chromatin accessibility and the transcriptome were minimally affected at 18 hpi, but changed dramatically by 4 dpi, these results indicate that SOX4 binding precedes major phenotypic changes.

## SOX4 targets genes for silencing and activation during reprogramming
We next sought to understand the mechanistic consequences of SOX4 binding. To assess the association of the ATAC peaks with gene expression, we annotated each peak with its nearest gene, associating 19,985 genes in total to all ATAC peaks ($n = 127,787$) identified at any of the newly opened, unchanged, or newly closed regions at 4 dpi. The annotated genes were then ranked based on the fold-change between empty vector and *Sox4*-expressing hepatocytes, and the ranked gene list was used as input for GSEA[39]. Consistent with our RNA-Seq results, regions exhibiting decreased chromatin accessibility upon *Sox4* expression were found in proximity to hepatocyte genes (Fig. 4a, upper), while regions exhibiting increased chromatin accessibility upon *Sox4* expression were found in proximity to reprogramming-associated genes (Fig. 4a, lower). Approximately half of the genes associated with newly closed regions (2685/5272; 50.9%) were down-regulated during DDC-induced reprogramming, including many canonical hepatocyte genes (Fig. 4b, Supplementary Table 1). By contrast, more than half of the genes associated with newly opened regions (3663/5823; 62.9%) were upregulated during DDC-induced reprogramming, including many known biliary genes (Fig. 4c, Supplementary Table 1). Gene ontology (GO) analysis of ATAC-Seq

(Supplementary Data 2) and RNA-Seq (Supplementary Data 3) data revealed that terms enriched for newly closed region-associated genes were shared with the terms under-represented in DDC-induced reprogrammed cells (Fig. 4d, e, Supplementary Fig. 12a, highlighted in blue letters), while terms enriched for newly opened region-associated genes were shared with the terms over-represented in DDC-induced reprogrammed cells (Fig. 4f, g, Supplementary Fig. 12b, highlighted in red letters). Applying a similar analytical approach to our SOX4 CUT&RUN-Seq data, when we annotated SOX4-bound peaks at the 18 hpi and 4 dpi timepoints with the nearest gene and performed GSEA, there was enrichment of a hepatocyte signature for genes associated with SOX4 peaks at 18 hpi (Fig. 4h, upper) and enrichment of a reprogramming signature for genes associated with SOX4 peaks at 4 dpi (Fig. 4h, lower). Taken together, these results indicate that *Sox4*-induced changes in chromatin accessibility are associated with corresponding decreases in the expression of hepatocyte genes and increases in the expression of biliary genes, mirroring the changes seen during DDC-induced reprogramming.

We then performed DNA footprint analysis to identify transcription factor binding sites whose footprints changed following *Sox4* expression. We compared these inferred transcription factor binding results to the enriched transcription factor motifs we previously identified in newly opened regions during DDC-induced reprogramming[27]. Notably, many of the transcription factors inferred to bind to newly opened regions following *Sox4* expression were consistent with those found in DDC-associated reprogramming motifs (e.g. AP1, E2F, and TEAD)[27] (Fig. 4i). Interestingly, we also found that several transcription factors associated with hepatocyte identity and function (e.g. HNF4α/γ, RARα, and CEBPα/β/δ/ε/γ) were less DNA-bound following *Sox4* expression (Fig. 4i). Notably, these results were consistent with our earlier observation that HNF4A motifs were present in many SOX4 CUT&RUN peaks (Fig. 3f). Collectively, these results indicate that changes in chromatin and transcription factor binding landscapes following *Sox4* ectopic expression resemble those of cells undergoing physiological (toxin-induced) biliary reprogramming.

## SOX4 initially closes active hepatocyte enhancers and evicts HNF4A
Strikingly, genes associated with newly closed regions (18 hpi) that were bound by SOX4 ($n = 825$ genes) exhibited a marked reduction in gene expression during DDC-induced reprogramming (Fig. 5a, right bottom). By contrast, genes associated with newly closed regions that were not bound by SOX4 ($n = 5090$ genes) exhibited much smaller reductions in gene expression (Fig. 5a, right top). We observed that newly closed regions were highly enriched in areas distal to the transcription start sites (TSS) (Supplementary Fig. 13a, b), raising the possibility that SOX4 binding modulates the activity of hepatocyte enhancers.

To assess the initial chromatin targeting by SOX4 more deeply, we compared sites of SOX4 binding with regions previously characterized based on their sensitivity to MNase and histone modifications. Upon high-level MNase digestion, labile nucleosomes and free DNA are destroyed and stable nucleosomes are resistant, whereas low-level MNase digestion preserves labile nucleosomes at enhancers[40] (Fig. 5b). Open regions targeted first by SOX4, that later became closed, initially exhibited MNase profiles resembling those of active liver enhancers (Fig. 5c, d) and had ChIP-Seq patterns of H2B and H3 (GSE57559[40]) similar to those associated with active hepatocyte enhancers (Fig. 5e, f, compare rows 2 and 4). Moreover, *Sox4* expression was associated with a statistically significant reduction in accessibility of these enhancers at 4 dpi compared to the empty vector control (Fig. 5g, h, Supplementary Fig. 13c) and resulted in a decrease of the active enhancer marks H3K27ac and H3K4me1 at 4 dpi compared to 18 hpi, with minimal change in repressive mark H3K27me3 (Fig. 5g, h, Supplementary Fig. 13d). To test the possibility that SOX4 may be involved

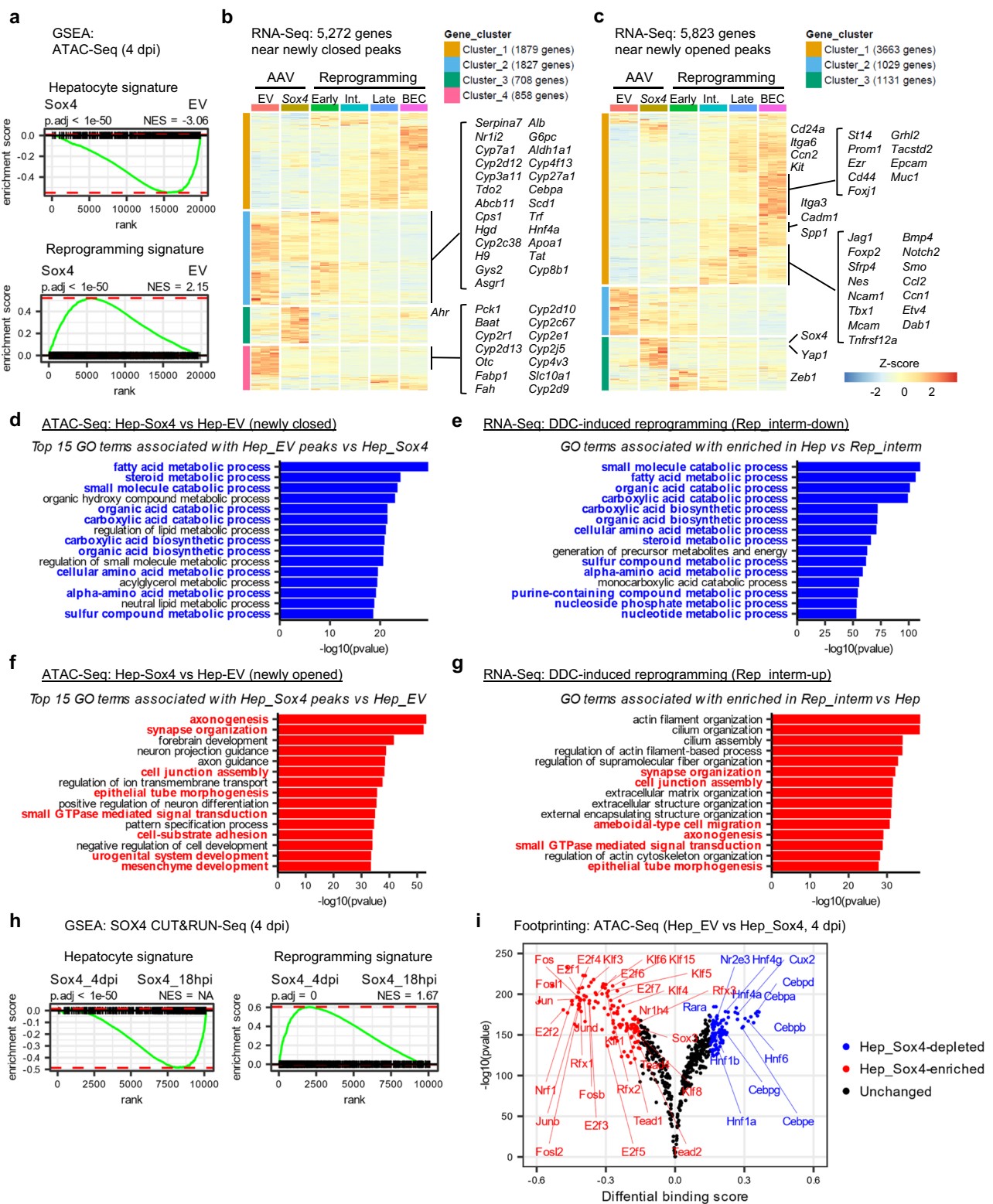

in histone deacetylation at hepatocyte enhancers, we asked whether ectopically expressed SOX4 in hepatocytes can directly interact with any of HDAC1/2/3, the main contributors to histone lysine deacetylation[41]. Co-immunoprecipitation assays demonstrated that SOX4 can directly bind HDAC1 and, to a lesser extent, HDAC3 (Supplementary Fig. 13e), suggesting that SOX4 may alter histone acetylation through the recruitment of HDAC proteins. Collectively, these results indicate that shortly after its induction, SOX4 binds to and inactivates active hepatocyte enhancers.

Next, we sought to understand how SOX4 targets hepatocyte enhancers. Given the unexpected finding that the HNF4A binding motif was enriched in both footprinting and CUT&RUN experiments (Figs. 3f and 4i), we considered the possibility that targeted rather than promiscuous binding was responsible for SOX4's localization to HNF4A sites. To this end, we first separated the SOX4 binding sites to those enriched for 18 hpi and those enriched for 4 dpi (Fig. 6a) and found that the HNF4A motif was the most enriched motif in the 18 hpi-specific SOX4 binding sites. We then realized that the SOX binding

**Fig. 4 | SOX4 binding silences hepatocyte genes while priming biliary genes by altering binding patterns. a** GSEA was performed with genes annotated to 4dpi ATAC-Seq peaks to compare empty vector and *Sox4* hepatocytes using the hepatocyte-enriched and Rep_early-enriched signatures (see Supplementary Data 1, Fig. 2j, Methods). Heatmap visualization of the 5,272 genes near newly closed peaks (**b**) and 5,823 genes near newly opened peaks **c** using the RNA-Seq data of the DDC-induced reprogramming experiment (Fig. 2i, j) (*n* = 3 animals). Genes were categorized to four (**b**) or three (**c**) clusters by *k*-means clustering. Representative hepatocyte (39/50) and biliary/reprogramming genes (33/49 from a manually curated gene list; Supplementary Table 1) are shown on the right. Newly closed (**d**) and opened peaks (**f**) in *Sox4* expressing hepatocytes at 4 dpi compared with empty vector hepatocytes (Fig. 3a) were annotated with the nearest genes, and this gene list was used as the input for GO enrichment analysis (*n* = 3 animals). GO analysis of RNA-Seq was also performed using genes downregulated (**e**) and upregulated (**g**) in Rep_intermediate cells vs Hep during DDC-induced reprogramming (*n* = 3 animals).

The same analysis comparing different stages, namely Rep_early vs Hep and Rep_late vs Hep are also shown in Supplementary Fig 12a, b. GO terms shared between the newly-closed and newly opened region-associated gene set and DDC-induced reprogramming context at any reprogramming stages are highlighted in bold blue (**d**, **e**) and red (**f**, **g**) texts, respectively. **h** GSEA was performed with genes annotated to the SOX4 CUT&RUN peaks identified either at 18 hpi and 4 dpi using the hepatocyte-enriched and Rep_early-enriched signatures as described earlier (Supplementary Data 1, Fig. 2j, Methods). **i** Global footprint analysis depicted as a volcano plot. The analysis used all motifs assigned as "bound" by TOBIAS in either empty vector or *Sox4* hepatocytes (*n* = 3 animals). Differential footprints are defined as those with Log2(fold-change of footprint score) >0.15 and log10(*p* value) < −100. Statistical testing was performed using the TOBIAS tool with the default setting. **a**, **h** Adjusted *P* values and NES values were calculated using the R fgsea package.

---

motif (CTTTGT/ACAAAG) partially overlaps the binding motifs for HNF4A (CAAAG/CTTTG) and other hepatocyte-enriched transcription factors (Fig. 6b). Further analysis revealed that footprints of HNF4A and RXRA, another hepatocyte transcription factor whose binding motif overlaps that of SOX4, were decreased in hepatocyte enhancers of *Sox4*-expressing hepatocytes (Fig. 6c), implying that SOX4 may compete with hepatocyte transcription factors to bind to these motifs. Moreover, through analysis of published ChIP-Seq data[42], we found substantial overlap of SOX4 binding sites with HNF4A binding sites in adult hepatocytes (Fig. 5h, magenta track on the bottom). These observations prompted us to hypothesize that SOX4 can specifically recognize and bind to the partial motif for HNF4A.

To directly investigate this hypothesis, we tested whether purified SOX4 protein can bind to an HNF4A motif using an in vitro electrophoretic mobility shift assay (EMSA) (Supplementary Fig. 14). We selected a motif located in a *Cyp2e1* distal enhancer, which was targeted by SOX4 and became less accessible at 4 dpi (Fig. 6d). We made double stranded DNA oligonucleotides of the wild-type (WT) site and of a mutant site with a scrambled HNF4A motif (Fig. 6d) and performed binding titrations (Fig. 6e). As predicted, recombinant HNF4A protein bound specifically to the wild type HNF4A site (Fig. 6e). Surprisingly, SOX4 bound the same site even more robustly than purified HNF4A, with marked specificity (Fig. 6e). Thus, SOX4 is able to recognize a specific HNF4A site, both in vivo and in vitro, embedded in a gene that is normally targeted by HNF4A.

Next, we performed a competition experiment to determine whether purified SOX4 protein can displace HNF4A. To this end, we first saturated the WT HNF4A-specific probe with 30 nM HNF4A, and then titrated SOX4 into the reaction. Strikingly, 10–30 nM SOX4 effectively displaced HNF4A protein from the probe (Fig. 6f). Collectively, these data suggest that SOX4 suppresses the hepatocyte identity in part by evicting resident hepatocyte transcription factors through competitive binding, thereby reducing the accessibility and activity of hepatocyte enhancers.

### SOX4 opens chromatin associated with biliary genes
We next turned our attention to the induction of the biliary program. Specifically, we explored the possible role of SOX4 as a pioneer factor by testing its ability to directly mediate the activation of repressed chromatin. Based on the strong correlation between SOX4 binding and changes in the chromatin landscape (Fig. 3i), we first assessed the ability of SOX4 to bind nucleosomal DNA, as this is a necessary feature of pioneer factor activity. To this end, we incubated recombinant SOX4 protein with nucleosome particles assembled on a human *LIN28B* DNA fragment, which has served as a standard for establishing the pioneer transcription factor activity of SOX2[16]. Consistent with our model, EMSA confirmed that recombinant SOX4 binds to assembled *LIN28B* nucleosomes at low nanomolar concentrations (Fig. 7a). Given that pioneer factor activity is associated with opening of previously closed

chromatin, we divided the regions of newly opened chromatin (based on the ATAC-Seq peaks) into two categories: those whose initial accessibility increased over baseline upon *Sox4* expression ("more opened regions," MORs) (Fig. 7b, upper left) and those regions that only became accessible when *Sox4* was expressed ("de novo opened regions," or DORs) (Fig. 7b, lower left). We visualized SOX4 CUT&RUN signals at these regions and confirmed that both MORs and DORs were bound by SOX4 at 4 dpi (Fig. 7b, right). DORs, and to a lesser extent MORs, fell into broad domains of chromatin enriched for high-MNase signals and were enriched for core histones (Fig. 7c–f, rows 1 and 2), compared to active promoters and enhancers (Fig. 7c–f, rows 3 and 4), demonstrating that the MOR and particularly DOR sites were originally in more compacted chromatin. We conclude that SOX4 acts as a pioneer factor by targeting nucleosomal DORs, and to a lesser extent MORs, leading to increased chromatin accessibility.

Beyond chromatin accessibility, we next considered how *Sox4* expression alters the landscape of a range of histone marks. Comparing 18 hpi and 4pi, at DORs, active enhancer and promoter marks were increased from initially faint or residual background levels, and at MORs the active marks were increased from baseline, while the H3K27me3 repressive mark was marginal at baseline and reduced at 4 dpi (Supplementary Fig. 15a). In accordance with our earlier finding that newly opened regions were enriched in proximity to reprogramming-related genes (Fig. 4a, c), comparable changes in histone marks were observed in the regions near biliary genes; concordantly, SOX4 binding was observed either at putative TSS-distal enhancers (Fig. 7g, left) or at promoters/gene bodies (Fig. 7g, right). Using the ATAC-Seq data from our earlier DDC-induced reprogramming study[27], we observed that MORs and DORs were indeed opened in reprogrammed and biliary cells (Fig. 7h, black tracks). Taken together, these results suggest that SOX4 binding at 4 dpi can lead to chromatin opening required for biliary reprogramming.

### SOX4 targeting predicts the chromatin changes seen in DDC-induced reprogramming
Lastly, we considered whether the consequence of ectopic *Sox4* expression in hepatocytes mirror the epigenetic changes seen in DDC-induced reprogramming. To this end, we profiled H3K27ac, H3K4me1, H3K4me3 and H3K27me3 at each reprogramming stage by Cleavage Under Targets and Tagmentation (CUT&Tag)-Seq[43,44], which allowed us to obtain reliable data with as few as 10,000–70,000 input cells (Supplementary Fig. 15b). We then mapped the CUT&Tag data to the SOX4 binding sites identified in the ectopic *Sox4* expression system. Strikingly, active histone marks gradually decreased in 18hpi-enriched ectopic SOX4 binding sites during DDC-induced reprogramming (Supplementary Fig. 15c, upper heatmaps; blue aggregate plots above the heatmaps). As DDC-induced reprogramming proceeded, active histone marks increased in 4dpi-enriched ectopic SOX4 binding sites (Supplementary Fig. 15c, lower heatmaps; green aggregate plots above

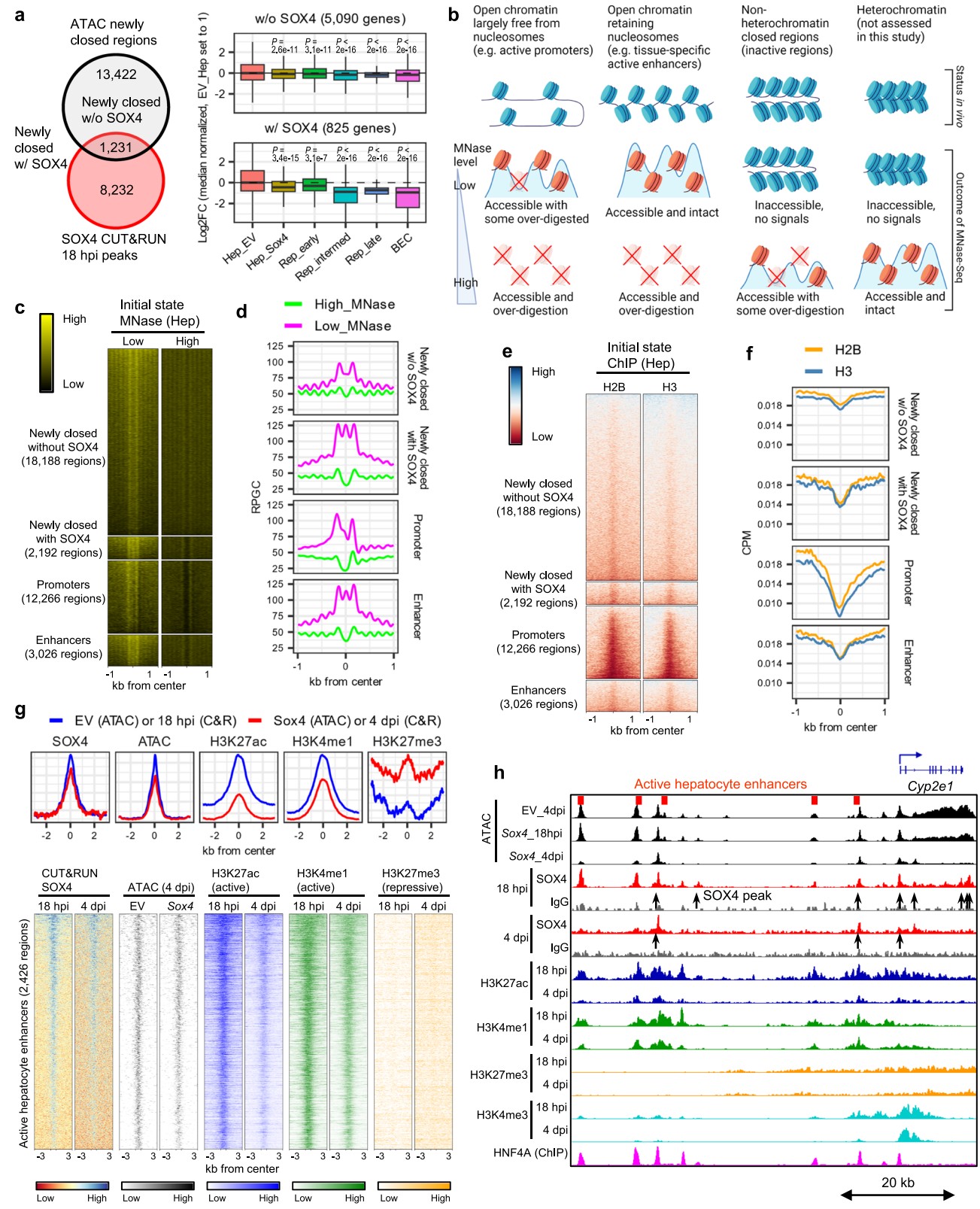

the heatmaps). Additionally, the repressive mark, H3K27me3 moderately declined during DDC-induced reprogramming at SOX4-bound regions at 4 dpi, but not 18 hpi (Supplementary Fig. 15c, lower heatmaps; green aggregate plots above the heatmaps). Together, the results support the hypothesis that *Sox4*-mediated alteration in epigenetic states in hepatocytes contribute to injury-induced biliary reprogramming by modulating active and repressive histone marks.

## Discussion

The epigenetic mechanisms underlying cell fate switching, or reprogramming, have been studied in detail in the context of induced pluripotency, whereby pioneer factors enable a change in cell identity by reconfiguring chromatin. Our study provides evidence that SOX4 acts as a pioneer factor in vivo in a validated system of physiological reprogramming: hepatobiliary metaplasia. Moreover, we found that SOX4

**Fig. 5 | SOX4 suppresses hepatocyte identity by inactivating hepatocyte enhancers. a** Newly closed regions with or without overlaps with 18hpi-SOX4 peaks were annotated with their nearest genes, and expression of each gene in empty vector or *Sox4*-expressing hepatocytes was compared with the DDC-induced reprogrammed cells. *P* values were calculated by Wilcoxon rank sum test with Hep_EV as the reference are shown without adjustment (*n* = 3 animals). The center line, box limits, whiskers, and points indicate the median, 25th/75th quartiles and 1.5× interquartile range, respectively. **b** Schematic representing the nucleosomal states and the corresponding outcomes of MNase-Seq nucleosomal states are roughly categorized into four groups, with representation of how they are detected by different levels of MNase treatment. **c** Heatmap representation of MNase-Seq at low and high levels for the newly closed regions with or without overlaps with SOX4 peaks at 18 hpi along with active liver promoters and enhancers. The corresponding

averaged aggregate plots are shown in (**d**). **d** Averaged aggregate plots of hepatocyte MNase-Seq in newly closed regions with or without overlaps with SOX4 peaks (18 hpi) along with active liver promoters and active liver enhancers. **e** Heatmap representation of ChIP-Seq for core histone H2B and H3 for the same regions as in (**c**). The corresponding averaged aggregate plots are shown in (**f**). **f** Averaged aggregate plots of H2B and H3 ChIP-Seq for the same regions as in **c**. **g** ATAC-Seq and CUT&RUN-Seq of SOX4, H3K27ac, H3K4me1 and H3K27me3 signals visualized as heatmaps at active liver enhancer regions. The corresponding averaged aggregate plots are shown on the top. Aggregate plots with y-axis values are shown in the Source Data File. **h** Genome browser view of the *Cyp2e1* gene is shown as an example of SOX4-induced closing of a hepatocyte gene-associated region. HNF4A ChIP-Seq data were obtained from GEO datasets (GSE137066) reflecting normal liver cells. **c**–**f** Analyses done using GEO dataset GSE57559.

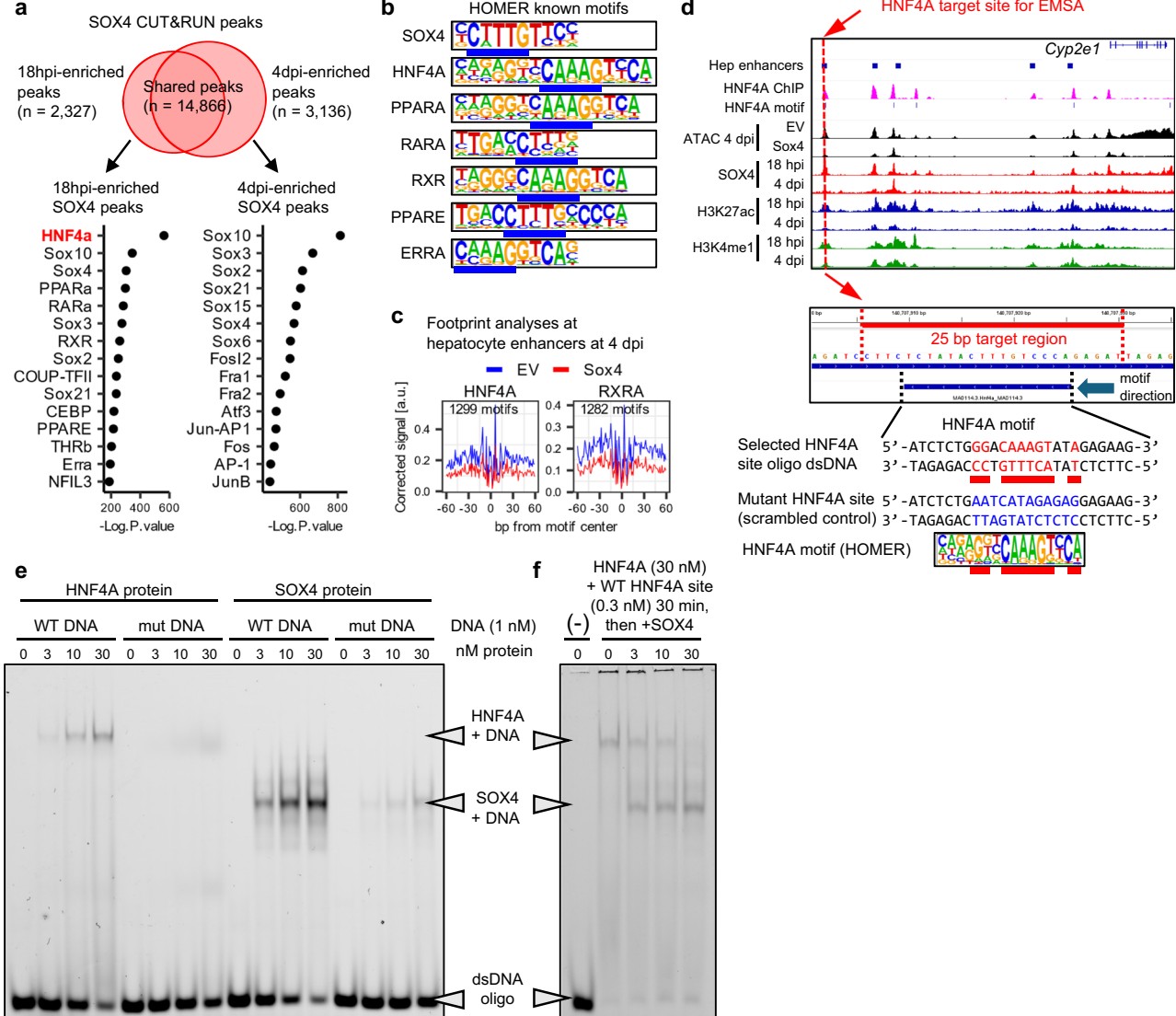

**Fig. 6 | SOX4 evicts hepatocyte transcription factors by hijacking their binding motifs. a**. HOMER motif analysis with known motifs for 18hpi- and 4dpi- enriched SOX4 CUT&RUN peaks. 18hpi-enriched SOX4 peaks (*n* = 2327) and 4dpi-enriched SOX4 peaks (*n* = 3136) were identified using DiffBind and DESeq2 packages (*n* = 3 animals). The HNF4A motif, highlighted in red, ranked first among 18 hpi-enriched SOX4 binding sites. **b** Consensus binding sequences of several hepatocyte transcription factors exhibiting partial overlap with that of SOX4 (overlapped sequences are underlined). **c** ATAC-Seq footprinting analysis of hepatocyte transcription factors HNF4A and RXRA at motifs overlapping the hepatocyte enhancers. The data are shown as the comparison between empty vector and *Sox4* hepatocytes (*n* = 3 animals). Motifs assigned as "bound" by TOBIAS for the empty

vector and *Sox4* expressing samples are combined and used for the analysis. **d** Schematic representing the *Cyp2e1* distal enhancer with an HNF4A motif for which a ds-DNA oligonucleotide and its scrambled mutant were designed. Most conserved sequence in the HNF4A motif is highlighted in red texts along with the consensus HNF4A motif sequence shown below. **e** EMSA was performed to assess the ability of recombinant mouse HNF4A protein and mouse SOX4 protein to bind the wild type and mutant *Cyp2e1* enhancer sequence. **f** EMSA was performed to assess whether SOX4 can displace HNF4A pre-bound to the wild type *Cyp2e1* enhancer dsDNA oligonucleotide by titrating SOX4 protein concentration used for competition. **e**, **f** A representative gel image from 2 replicates is shown.

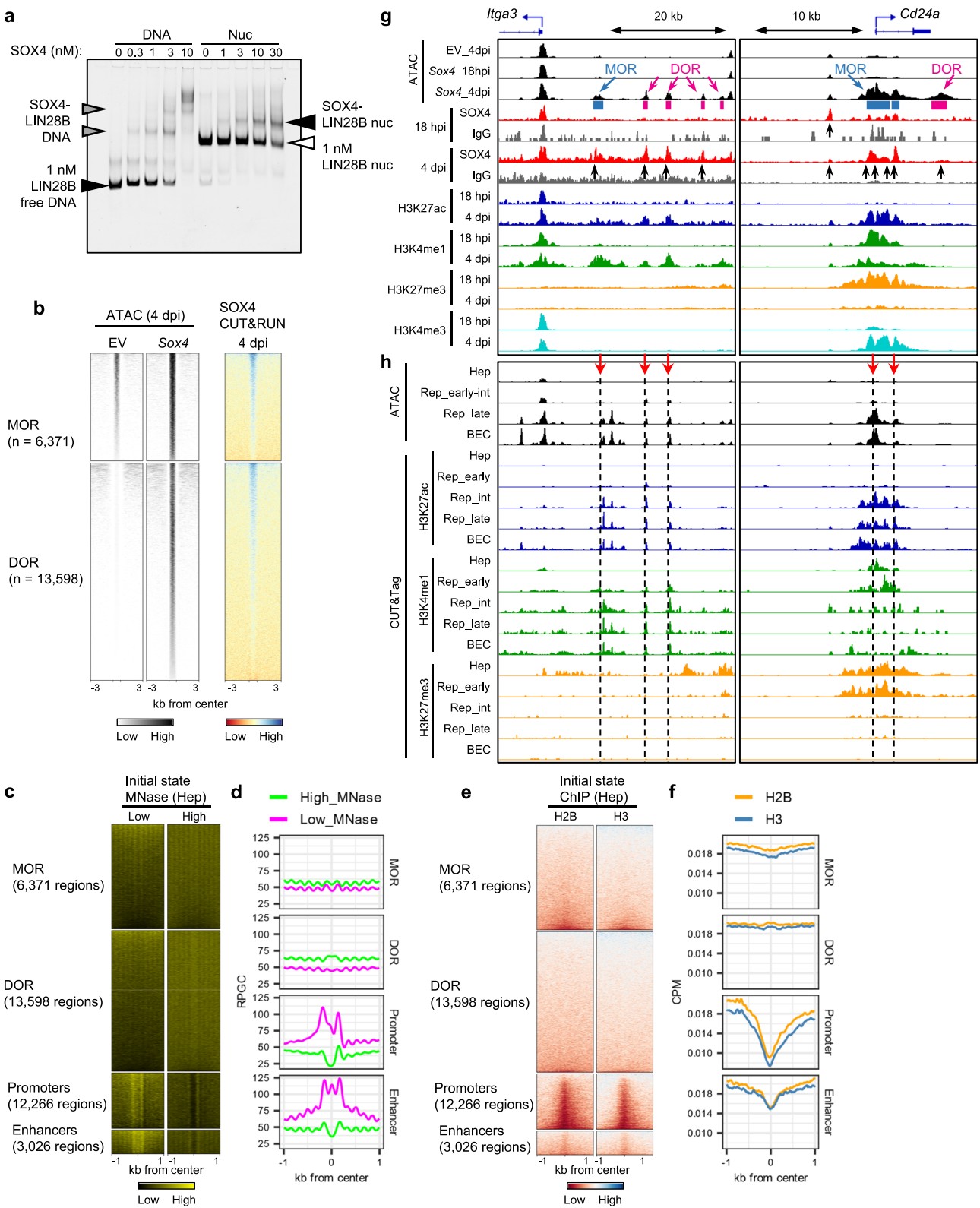

exerts this activity through a sequential process in which enhancers associated with the starting (hepatocyte) cell type are decommissioned prior to the activation of enhancers associated with the acquired (biliary) cell type. Such enhancer reorganization by pioneer factors, including silencing of the starting cell's enhancers, has been proposed for iPSCs, where early and preferential binding to active somatic enhancers causes the redistribution of associated somatic transcription factors like P300 and the recruitment of enhancer silencing factors like

HDAC1[13]. Our study also confirmed the direct association of SOX4 and HDAC1 via their protein-protein interactions. We expect future studies will further investigate how the distribution of HDAC1 and other potential HDACs are altered by SOX4. In addition, pioneer factors can compete for binding sites in enhancers to influence lineage trajectories[45,46]. Whereas prior studies investigated cellular reprogramming in culture, our study defines how powerful reprogramming transcription factors can be in the context of a tissue, with many more

**Fig. 7 | SOX4 opens putative biliary cell enhancers in hepatocytes. a** EMSA was performed to evaluate the binding ability of recombinant mouse SOX4 to naked *LIN28B* DNA and in vitro assembled nucleosomal *LIN28B* DNA. A representative gel image from 2 repeated experiments is shown. **b** SOX4 CUT&RUN-Seq data (right column) are shown for the more opened regions (MOR) and de novo opened regions (DOR), which correspond to newly opened regions with weak ATAC peaks in empty vector and those without ATAC peaks in empty vector (left two columns). **c** Heatmaps of previously published MNase-Seq data (GSE57559) obtained for adult hepatocytes at low and high levels of MNase[40]. The regions are centered with all the intersectable ATAC peaks of either empty vector or *Sox4*-expressing hepatocytes. **d** Averaged aggregate plots corresponding to the heatmaps shown in **c. e** Heatmaps of previously published H2B and H3 ChIP-Seq obtained for adult hepatocytes

(GSE57559)[40]. Regions are centered with all intersectable ATAC peaks of either empty vector or *Sox4* hepatocytes. **f** Averaged aggregate plots corresponding to the heatmaps shown in (**e**). **g** Genome browser views of two examples for *Sox4*-induced opening at biliary epithelial cell genes: *Itga3* (intermediate-to-late reprogramming marker) and *Cd24a* (early-to-intermediate reprogramming marker). **h** Genome browser views of ATAC-Seq and CUT&Tag-Seq data in the same regions as **g**. ATAC-Seq data of reprogrammed cells and biliary cells are adapted from our earlier study[27]. Hep ATAC-Seq data indicates the Hep_EV data collected in this study, which were downsampled to adjust the read depth comparable to the DDC-induced reprogrammed and biliary cells (as shown in Fig. 3). CUT&Tag-Seq data were collected for each stage of DDC-induced reprogramming.

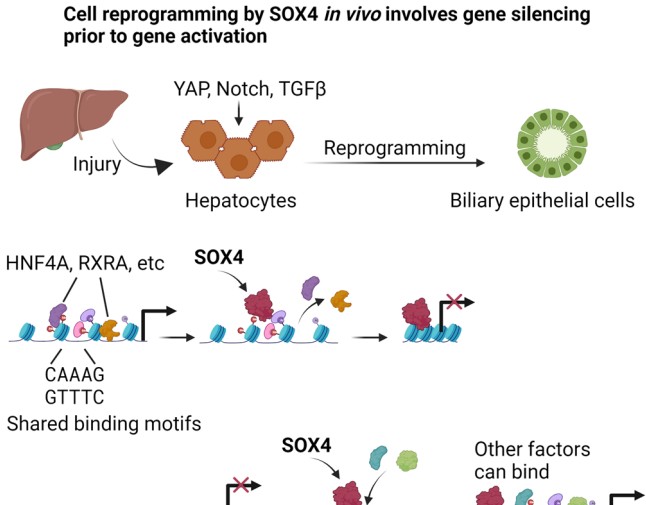

**Cell reprogramming by SOX4 *in vivo* involves gene silencing prior to gene activation**

**Fig. 8 | Proposed model of the SOX4-mediated regulatory mechanism of hepatobiliary reprogramming.** Liver injury activates signaling pathways which are responsible for induction of biliary reprogramming, such as YAP, Notch and TGFβ pathways. *Sox4* expression is readily activated as one of the earliest responses to these pathways. Upon translation, SOX4 binds hepatocyte enhancers through recognition of the shared motif sequences, and evicts the originally bound hepatocyte transcription factors, including HNF4A and RXRA, which leads to repression of hepatocyte phenotypes. Then, SOX4 starts to bind closed and silenced chromatin regions required for acquisition of biliary phenotypes and open these regions as a traditional pioneer factor to establish access for other factors to these regions.

stimuli such as soluble factors, basement membranes, and cell-cell contacts that normally help restrain cell fate changes from occurring.

Our findings offer a molecular mechanism to account for the specificity of binding in a physiologically relevant context. Specifically, we observed that key hepatocyte transcription factors – including HNF4A, a master regulator of hepatocyte identity[47,48] – share binding motifs with SOX4. The finding that SOX4 can associate with a canonical HNF4A binding site more strongly than HNF4A itself is consistent with the lack of expression of *Sox4* in healthy hepatocytes and supports the notion that SOX4 hijacks HNF4A sites when it becomes expressed during injury-induced reprogramming. Consistent with this idea, SOX4 preferentially binds to open chromatin regions occupied by HNF4A early in biliary reprogramming. Subsequently, owing to the biochemically stronger binding affinity of SOX4, HNF4A was evicted from its native binding sites. Thus, our study provides evidence that pioneer factors can dually disrupt a starting cell fate while engaging closed chromatin to initiate a new cell fate. Meanwhile, the mechanism by which SOX4 adopts the role of a transcription activator or repressor is unclear. One possible hypothesis is that post-transcriptional modification of SOX4 could change its role, which we expect to be tested in future studies.

The results presented here support a multi-step model in which pioneer factor activity, through both positive and negative effects on

chromatin accessibility, is essential for the cell fate changes accompanying tissue metaplasia in vivo. By synthesizing our findings with those of earlier studies[23–27], we propose a model (Fig. 8) in which signals from the injured liver microenvironment (e.g. YAP, Notch, TGFβ pathway[23,25,26]) result in the induction of SOX4, which then binds to regions of open chromatin occupied by hepatocyte-specific transcription factors (e.g. HNF4A and RXRA), causing their displacement. Then, SOX4 acts as a traditional pioneer factor by binding to canonical SOX binding sites in biliary enhancers in closed chromatin, causing the regions to become open and accessible to other biliary transcription factors.

Meanwhile, our study indicates that *Sox4*-mediated reprogramming is restricted to the early-to-intermediate stages, and that other transcription factors including *Sox9* and *Hnf1b* are required for completion of reprogramming. Moreover, given that necessity for *Sox4* was weak in the loss-of-function study, pioneer activity of *Sox4* can be compensated for by other transcription factors. We did not observe compensatory upregulation of *Sox11* or *Sox12* (other *SoxC* family members) in *Sox4*-KO hepatocytes by RNA-Seq (data not shown). Therefore, certain transcription factors other than *Sox* family members could have compensated for the *Sox4* loss. Thus, further studies are expected to elucidate the transcriptional regulatory mechanisms underlying the transitions to the later stages of the metaplastic process, as well as the compensatory mechanism for pioneer activity of *Sox4*. Likewise, additional studies are needed to link extracellular signals to the activation of pioneer factor-mediated initiation of epigenetic cascades, such as recently seen for Kupffer cell-derived IL-6 initiating epigenetic changes in hepatocytes as they transition to biliary cells in the DDC-treated liver[49]. Finally, further technical developments in epigenomic profiling are needed to overcome challenges in transcription factor profiling. For example, HA-tagged SOX4 was necessary for detection via CUT&RUN sequencing; native SOX4 capture in injury-induced cells was not possible despite extensive protocol optimization.

Integration of the genomic, epigenomic, and biochemical pathways involved in this metaplastic cascade provides a foundation for the complex processes of cellular metaplasia in vivo. This will lead to a framework to better understand the core cellular processes underpinning physiological metaplastic processes in response to injury as well as deleterious pathophysiological metaplastic processes such as those leading to cancer[2].

## Methods
### Lead contact
Further information and requests for resources and reagents should be directed to and will be fulfilled by the lead contact, Ben Z. Stanger, email: bstanger@upenn.edu

### Materials availability
All unique and stable reagents generated in this study of which sufficient quantities exists are available from the Lead Contact with a completed Materials Transfer Agreement.

## Mice

Rosa-LSL-Cas9-EGFP mice[31] on a C57BL/6J background were purchased from the Jackson Laboratory (strain #026175) and maintained as homozygotes. All mouse experiment procedures used in this study were performed following the NIH guidelines. All mouse procedure protocols used in this study were in accordance with, and with the approval of, the Institutional Animal Care and Use Committee of the University of Pennsylvania.

For characterization of the reprogramming stage, 4-5 week-old mice were retro-orbitally injected with AAV:ITR-U6-sgRNA(backbone)-TBG-Cre-WPRE-hGHpA-ITR[30] (empty vector: EV) with $5 \times 10^{11}$ genome copies/mouse/gene. One week later, induction of biliary reprogramming was started by initiating a 0.1% 3,5-diethoxycarbonyl-1,4-dihydrocollidine (DDC) diet (Envigo). Eight weeks after the DDC challenge, reprogrammed cells and biliary cells were harvested as described below.

For the exogenous expression of *Sox4*, *Sox9* and *Sox4/Sox9*, 4- to 8-week-old mice were retro-orbitally injected with $5 \times 10^{11}$ genome copies/mouse/gene. For the *Sox4* expression experiments for RNA-Seq, ATAC-Seq, and CUT&RUN-Seq experiments, $1 \times 10^{12}$ genome copies/mouse of AAV8-TBG-*HA-Sox4*-P2A-*Cre* were retro-orbitally injected.

## Plasmid cloning

All PCR reactions were performed using the Phusion Flash High-Fidelity PCR Master Mix (Thermo) following the manufacturer's instructions. For all AAV plasmids, endotoxin was eliminated by treating the plasmids with Endozero columns (Zymo Research) before proceeding to AAV production. Transformation was performed using Stbl3 bacteria (Thermo) following the manufacturer's instruction.

**AAV-HA-Sox4-P2A-Cre plasmid.** Mouse *HA-Sox4-P2A* and *P2A-Cre* blocks were PCR-amplified from pLVXT-Sox4 (Addgene, #101121) and AAV:ITR-U6-sgRNA(backbone)-TBG-Cre-WPRE-hGHpA-ITR[30] respectively, using the primers listed in Supplementary Table 2. The AAV-TBG backbone was prepared by removing EGFP from pAAV.TBG.PI.eGFP.WPRE.bGH (Addgene, #105535) using NotI-HF (NEB) and BamHI-HF (NEB). The two PCR-amplified DNA blocks were inserted into the linearized AAV-TBG vector by Gibson Assembly using an NEBuilder HiFi DNA assembly kit (NEB) following the manufacturer's instruction.

**AAV-HA-Sox9-P2A-Cre plasmid.** Mouse *HA-Sox9-P2A* and *P2A-Cre* blocks were PCR-amplified from pWPXL-Sox9 (Addgene, #36979) and AAV:ITR-U6-sgRNA(backbone)-TBG-Cre-WPRE-hGHpA-ITR[30] respectively, using the primers listed in Supplementary Table 2. The two blocks were inserted into the linearized AAV-TBG vector prepared as described above using the NEBuilder assembly kit.

**AAV-FLAG-Sox4-P2A-Cre plasmid.** Mouse *FLAG-Sox4-P2A* and *P2A-Cre* blocks were PCR-amplified from pLVXT-Sox4 and AAV:ITR-U6-sgRNA(backbone)-TBG-Cre-WPRE-hGHpA-ITR respectively, using the primers listed in Supplementary Table 2. The two blocks were inserted into the linearized AAV-TBG vector prepared as above using the NEBuilder assembly kit.

**CMV-FLAG-Sox4 plasmid.** The *FLAG-Sox4* block was amplified from the AAV-*FLAG-Sox4-P2A-Cre* plasmid using a forward primer: 5'-AGTGCTAGCGCCACCATGGACTACAAAGACG, and a reverse primer: 5'- TCGTGTACATCAGTAGGTGAAGACCAGGTTAGAGATGC. The DNA block was then digested with NheI-HF (NEB) and BsrGI-HF (NEB) and cloned into the CMV backbone vector prepared by linearizing the mEGFP-N1-YAPS127A-L318E plasmid (Addgene, #166465) using NheI-HF and BsrGI-HF.

## AAV preparation

90–100% confluent 293T cells in 15 cm dishes were replenished with 15 ml fresh DMEM (Thermo) supplemented with 2% FBS (Thermo) without antibiotics. For a 15 cm plate, 16 μg AAV8-Rep/Cap plasmid (Grompe Lab), 16 μg Ad5-Helper plasmid (Grompe Lab), 16 μg AAV transfer vector, and 144 μl of 1 mg/ml polyethylenimine (PEI) (Polysciences) were mixed in 9 ml OptiMEM (Thermo). After incubation at room temperature (RT) for 15 min, plasmid/PEI complex was added to 293T cells in a dropwise manner, and the plates were gently rocked to mix. After incubation in a $CO_2$ incubator for 6 days, the cells and culture supernatant were harvested into 50 ml tubes and centrifuged at $1900 \times g$ for 15 min. The supernatant was transferred to new tubes, and 1/40,000 volume of Benzonase (Sigma-Aldrich) was added and mixed thoroughly by inversion. After digestion of non-viral DNA by incubating at 37 °C for 30 min, virus medium was centrifuged at $1900 \times g$ for 15 min, and the supernatant was filtered with a 0.22 μm a filter unit containing a PES membrane (Thermo). Then, 1/4 volume of 40% polyethylene glycol 8000 (PEG8000) in 2.5 M NaCl was added and mixed thoroughly by inversion. Following overnight incubation at 4 °C, precipitated AAV was collected by centrifugation at $3000 \times g$ for 15 min. After removal of the supernatant, precipitate was homogenized in 100 μl PBS per 15 cm dish by through pipetting. Non-AAV precipitate was eliminated by centrifugation at $2200 \times g$ for 5 min. Smaller debris were further removed by filtrating the eluted AAV with a 0.45 μm filter columns (Corning). This crude AAV was titrated by qPCR using the AAV8-TBG-Cre (Penn Vector Core) as a standard and a forward primer: 5'- GGAACCCCTAGTGATGGAGTT, and a reverse primer: 5'- CGGCCTCAGTGAGCGA and directly used for KO experiments without further purification.

## Immunofluorescence

Frozen sections were used for immunofluorescence. Tissue was fixed in zinc-formalin overnight, equilibriated in 30% sucrose/PBS, embedded in Tissue-Tek® O.C.T. compound (Sakura), and 8 μm sections were prepared. Remnant O.C.T. compound was removed by submerging the slides in PBS for 5 min. The specimens were permeabilized with 0.1% Triton X-100 (Fisher) in PBS at RT for 15 min. After treatment with the Blocking One Histo (Nacalai) at RT for 10 min, the specimens were incubated with primary antibodies (Supplementary Table 3) diluted in 1/20× Blocking One Histo at RT for 1 h or at 4 °C overnight. The sections were then stained using donkey anti-rabbit, rat, or goat antibodies conjugated with AlexaFluor 488, AlexaFluor 594 or AlexaFluor 647 (Invitrogen) (Supplementary Table 3) at 1/300 dilution and DAPI (Thermo) at 1/1000 dilution. After incubation at RT for 1 h, the specimens were mounted in Aqua-Poly/Mount (Polysciences), and imaged using an Olympus IX71 inverted fluorescent microscope.

## Hematoxylin and Eosin (H&E) staining

H&E staining was performed by Penn Molecular Pathology and Imaging Core (MPIC).

## Hepatocyte isolation

Livers were perfused with 40 ml of HBSS (Thermo), followed by 40 ml HBSS with 1 mM EGTA (Sigma), then 40 ml HBSS with 5 mM $CaCl_2$ (Sigma) and 40 μg/ml liberase (Sigma). Following perfusion, livers were mechanically dispersed with tweezers, resuspended in 10 ml wash medium (DMEM supplemented with 5% FBS), and filtrated with a 70 μm cell strainer. The cells were centrifuged at $50 \times g$ at 4 °C for 5 min. Then, the cells were resuspended in complete percoll solution (10.8 ml percoll (Cytiva), 12.5 ml wash medium, and 1.2 ml 10× HBSS per liver) and centrifuged at $50 \times g$ at 4 °C for 10 min. After a single wash with 10 ml medium, cells were spun at $50 \times g$ at 4 °C for 5 min and then used for downstream experiments.

## Whole liver cell isolation from normal mice

Livers were digested by the two-step liberase perfusion as described above. Then, the undigested remaining tissue was transferred to a 1.5 ml tube, minced with surgical scissors, and further digested with 10× concentrated liberase (~430 μl/tube of 400 μg/ml in HBSS with 5 mM CaCl$_2$) at 37 °C for 30 min while vortexing the sample several times intermittently. The digested tissue was filtered with a 70 μm cell strainer and combined with the cell suspension digested previously. The cells were then centrifuged at 300 × $g$ at 4 °C for 5 min. Then, the cells were suspended in 10 ml ACK lysis buffer (Quality Biological) and incubated on ice for 10 min to remove red blood cells. The cells were then collected by centrifugation at 300 × $g$ at 4 °C for 5 min and used for downstream analyses.

## Whole liver cell isolation from DDC-treated mice

Livers were digested by the two-step liberase perfusion as described above. Following perfusion, livers were submerged in 10 ml fresh HBSS with 5 mM CaCl$_2$, 40 μg/ml liberase and 40 μg/ml DNaseI (Millipore) in a C-tube (Miltenyi) and further digested using a gentleMACS Octo dissociator (Miltenyi) with a heating unit using the "37C_m_LIDK_1" protocol. Dissociated tissue was diluted in flow buffer (HBSS, pH 7.4) supplemented with 25 mM HEPES (Thermo), 5 mM MgCl$_2$ (MedSupply Partners), 1× Pen/Strep (Thermo), 1× Fungizone (Thermo), 1× NEAA (Thermo), 1× Glutamax (Thermo), 0.3% glucose (Sigma), 1× sodium pyruvate (Thermo) supplemented with 40 μg/ml DNaseI (hereafter flow buffer(+)). Undigested tissue was removed by passing it through a 70 μm cell strainer, and the cells were centrifuged at 300 × $g$ at 4 °C for 5 min. Then, the cells were suspended in 10 ml ACK lysis buffer and incubated on ice for 10 min. The cells were then collected by centrifugation at 300 × $g$ at 4 °C for 5 min and used for downstream analyses.

## Flow cytometry

Cells were resuspended in 2–3 ml flow buffer(+), and filtered with a 35 μm cell strainer equipped with a FACS tube (BD). The cell suspension was then transferred to a round-bottom 96 well plate at 100–150 μl/well and centrifuged at ~800 × $g$ at 4 °C for 1 min with a slow brake. The cells were then resuspended in 100 μl/well of flow buffer(+) containing fluorophore-conjugated antibodies (Supplementary Table 3) and incubated on ice for 20 min. After two washes in flow buffer(+) (150–200 μl/well, ~800 × $g$ at 4 °C for 1 min, slow brake), the cells were resuspended in flow buffer(+) containing 1/1000× TO-PRO-3 (Thermo) and analyzed using an LSR II flow cytometer (BD).

## Fluorescence-activated cell sorting (FACS) of DDC-treated whole liver cells

Cells were resuspended in 5 ml flow buffer(+) by centrifugation at 300 × $g$ at 4 °C for 5 min. After removal of the supernatant, the volume was increased to 1–1.5 ml with flow buffer(+), and 1/100 volume of rat anti-Cd45, rat-anti-Cd11b and rat anti-Cd31 antibodies (Supplementary Table 3) were added and incubated on ice for 10 min. After washing with 2 ml flow buffer(+) (300 × $g$ at 4 °C for 5 min, slow brake), the cells were resuspended in 5 ml flow buffer(+), 600 μl Dynabeads-anti-rat IgG (Thermo) were added, and the cell/bead mixture was incubated at 4 °C for 30 min with gentle tilting and rotation. The suspension was transferred to 5 ml FACS tubes and placed on a DynaMag™–5 Magnet (Thermo) for 2 min. The supernatant was transferred to a new tube, and the cells were collected by centrifugation at 2000 rpm (~800 × $g$) at 4 °C for 2 min with a slow brake. The cells were then resuspended in MACS buffer (PBS, 0.5% BSA, 2 mM EDTA) to the final volume of approximately 1.5 ml, and 150 μl CD326 (EpCAM) MicroBeads (Miltenyi) were added. After incubation at 4 °C for 15 min, the cells were washed with an equal volume of MACS buffer then centrifuged at 2000 rpm (~800 × $g$) at 4 °C for 2 min with a slow brake. Cells were resuspended in 2 ml MACS buffer, and Epcam+ cells and Epcam- cells were separated using LS columns (Miltenyi) following the manufacturer's instruction (4 columns were used per animal; 0.5 ml suspension/column). The cells were then collected by centrifugation at 2000 rpm (~800 × $g$) at 4 °C for 2 min with a slow brake. The cells were then resuspended in 0.5–1 ml flow buffer(+), and approximately 10–15 μl of each were set aside for fluorescence-minus one (FMO) controls (FMO-Brilliant Violet 421 (BV421): all stained except BV421-CD24; and FMO-PE/Dazzle 594: all stained except PE/Dazzle594-EPCAM), and stained in a 96 well round bottom plate as described earlier. The cells to be used for FACS were stained with BV421-CD24 (Biolegend), PE/Dazzle594-EPCAM (Biolegend), PE/Cy7-CD11b/CD31/CD45 (Supplementary Table 3) at 1:100 dilution in 15 ml tubes on ice for 20 min. After washing in 2 ml flow buffer(+) once by centrifugation at 2000 rpm (~800 × $g$) at 4 °C for 2 min with a slow brake, the cells were resuspended in 1–3 ml flow buffer(+) with 1/1000 TO-PRO-3, and the cells were sorted on an Aria II sorter (BD).

## Total RNA isolation and reverse transcription

Total RNA was extracted using the NucleoSpin RNA Kit (Takara) following the manufacturer's instructions. Approximately 500 ng of RNA was reverse transcribed in 20 μl volume using High Capacity cDNA Reverse Transcription Kit (Thermo). cDNA was diluted at 1:20 ratio in water and used for qPCR.

## qPCR

qPCR was performed at 10 μl/well using the Bio-Rad CFX 384 qPCR machine (Bio-rad). Each well contained: 3 μl diluted cDNA, 0.25 μl each of 10 μM forward and reverse primers (Supplementary Table 4), 1.5 μl H$_2$O and 5 μl SsoAdvanced SYBR reagent (Bio-rad).

## RNA-Seq

Library preparation and sequencing were performed by Novogene (Sacramento, CA) using a Novaseq 6000 (Illumina).

## Bioinformatics for RNA-Seq

Reads were aligned to the mouse genome (GRCm39) using STAR aligner with default parameters[50]. Gene-count matrices were produced by featureCounts[51]. To compare gene expression between samples, expression levels were normalized based on the "median of ratios" method using DESeq2[52]. To compare expression levels between different genes, "Transcripts per million (TPM)" normalization was performed using Salmon[53]. To build the gene sets for gene set enrichment analysis (GSEA), differential gene expression analysis was performed between hepatocytes and Rep_early cells using the median normalized data with the cut-off values of p.adj <0.05 and |log2(fold-change)| ≥ 1. The generated gene sets are listed in Supplementary Data 1.

## ATAC-Seq

50,000 cells were isolated from three AAV8-EV- or AAV8-*HA-Sox4*-P2A-*Cre*-injected (1 × 10$^{12}$ gc/mouse) liver samples at 18 hpi and 4 dpi and used as input for ATAC-Seq library preparation. Libraries were prepared as described[34] with minor modifications. Briefly, nuclei were isolated from the cells using a solution of 10 mM Tris-HCl pH 7.4, 10 mM NaCl, 3 mM MgCl$_2$, 0.1% IGEPAL CA-630. Immediately following isolation, the transposition reaction was conducted using Tn5 transposase (Diagenode) and TD buffer (Illumina) for 30 min at 37 °C. Transposed DNA fragments were purified using a Qiagen MinElute Kit, barcoded, and PCR amplified for 7–9 cycles depending on the samples using NEBNext High Fidelity 2× PCR master mix (New England Biolabs). The optimal cycle number was determined empirically each time by qPCR. The libraries were then purified with AMPure XP beads. Paired-end 150 × 2 sequencing was performed by Novogene (Sacramento, CA) using a NovaSeq 6000 (Illumina).

## Bioinformatics for ATAC-Seq

Reads were aligned to the mouse genome (mm10) using Bowtie2[54] with options "--very-sensitive -X 1000 --dovetail −1", and duplicates were removed using Picard (http://broadinstitute.github.io/picard/). Peak calling was performed using MACS2[55] with an FDR of 0.01 (default setting). Motif analysis was performed using HOMER (http://homer.ucsd.edu/homer/motif/) with the option "-size 300 −mask". Differential peak analysis was performed using triplicate samples (BAM files and peaks called independently for each replicate) using DiffBind[56] and DESeq2. Differential peaks were then annotated to the nearest genes using the annotatePeak function of ChIPSeeker R package[57], and the expression of these genes during DDC-induced reprogramming was analyzed using the RNA-Seq data as described above (GSE218945). Using the genes enriched in the newly closed/opened regions, gene ontology analysis was performed using the "enrichGO" function in clusterProfiler R package[58]. Analysis of the genomic distribution of the ATAC peaks was performed using plotAnnoBar and plotDistToTSS functions in ChIPSeeker. For GSEA, the annotated genes were ranked based on the log-fold change between EV and *Sox4*-expressing hepatocytes calculated by DESeq2, and the ranked gene list was used as input for the fgsea R package (fast preranked GSEA)[39]. For visualization using the Integrative Genomics Viewer (IGV) track browser[59] and deepTools[60] (for generation of heatmaps), BAM files were converted to bigwig files using bamCoverage with the "reads per genome coverage (RPGC)" normalization method.

Footprinting analysis was performed on replicate-merged BAM files using TOBIAS[35] with the default settings. For global foot printing analysis, all the foot prints that were assigned "bound" either in EV_Hep or Sox4_Hep were used as the input for the analysis (default setting of the TOBIAS pipeline). The output "bindetect_results" file was imported to R for visualization. For visualization of the aggregate footprints, corrected bigwig signals were retrieved using the ScoreBed function, and plotted using the ggplot2 R package. TOBIAS scores were retrieved using the "PlotAggregate" function with the option, "-output-txt", and visualized using ggplot2.

## CUT&RUN-Seq

CUT&RUN DNA was prepared following EpiCypher® CUTANA™ CUT&RUN Protocol v2.0 with minor modification. Briefly, 500,000 cells were isolated from AAV8-*HA-Sox4*-P2A-*Cre*-injected ($1 \times 10^{12}$ gc/mouse) livers at 18 hpi and 4 dpi ($n = 3$, each timepoint). The cells were washed in wash buffer (20 mM HEPES, pH 7.5, 150 mM NaCl (Sigma), 0.5 mM spermidine (Sigma) and EDTA-free protease inhibitors (Roche)) twice followed by centrifugation at $600 \times g$ at 4 °C for 3 min. The cells were resuspended in 150 µl wash buffer, 15 µl Concanavalin A-coated magnetic beads (EpiCypher) were added, and the cells/bead conjugate was bound to a magnet. After removal of the supernatant, the cells were resuspended in 50 µl antibody reaction buffer (wash buffer with 5% digitonin and 0.5 M EDTA), 0.5 µl antibody was added (Supplementary Table S3) and incubated at 4 °C overnight. Following two washes in 200 µl permeabilization buffer (wash buffer with 5% digitonin), beads were resuspended in 50 µl permeabilization buffer. Then, 2.5 µl pAG-MNase (EpiCypher) was added, and the samples were incubated at RT for 10 min. While on the magnet, the supernatant was removed, and the samples were washed twice in 200 µl cold permeabilization buffer. Following resuspension in 50 µl permeabilization buffer, 1 µl 100 mM CaCl$_2$ was added to activate MNase, and MNase digestion was performed at 4 °C for 2 h. The reaction was stopped by adding 33 µl STOP buffer (340 mM NaCl, 20 mM EDTA, 4 mM EGTA, 10 µg/ml RNase A, 50 µg/ml glycogen), and 1 µl E. coli spike-in DNA (0.5 ng/µl) (EpiCypher) to each tube. After incubation at 37 °C for 10 min, beads were bound to the magnet, and the supernatant was transferred to a new tube. The DNA was cleaned up with the NEB Monarch kit (NEB), eluted in 12 µl elution buffer, and used as CUT&RUN DNA. Library preparation was performed using NEBNext

Ultra II End Prep kit (NEB) with a slightly modified protocol. Briefly, after adaptor ligation, HA-Sox4 CUT&RUN DNA samples were purified with 1.75× AMPure XP beads (Beckman), while histone and IgG CUT&RUN samples were purified with 1.1× AMPure XP beads. 14-cycle PCR with index primers (NEB) was performed (initial denaturation: 1 cycle of 98 °C for 45 s; annealing/extension: 14 cycles of 98 °C for 15 s and 60 °C for 10 s; final extension: 72 °C for 1 min). Finally, the libraries were purified with AMPure XP beads (for HA-Sox4 0.8× followed by 1.2×; for histone and isotype control 0.9×). For samples with adaptor contamination, further AMPure bead cleanup or gel extraction was performed. Sequencing was performed using an Illumina NextSeq500 with a 150 cycle mid-output reagent kit (75-bp paired end) and an Illumina NextSeq2000 with 100 cycle S2 reagent kit (65-bp paired end).

## Bioinformatics for SOX4 CUT&RUN-Seq

SOX4 and isotype control CUT&RUN-Seq data were obtained in two separate sequencing experiments. The 75-bp data obtained from the NextSeq500 experiment were first trimmed with Cutadapt (ver. 4.1) to shorten the reads to 65-bp in order to merge into single 65-bp fastq files. Reads were aligned to the mouse genome (mm10) using Bowtie2 with the option "-X 1000", and duplicates were removed using Picard. For SOX4 differential peak analysis, peak calling was performed for each of the three biological replicates using MACS2 with each of the SOX4 BAM files and the combined isotype control BAM file for each timepoint. The FDR was set to 0.1 for peak calling of individual samples, and the called peaks showed robust enrichment of SOX binding motifs as confirmed by HOMER. This analysis generated $13,081 \pm 5151$ and $28,160 \pm 13,294$ peaks for 18 hpi and 4 dpi samples, respectively. Using these peaks and BAM files for each replicate, differential peak analysis was performed using DiffBind and DESeq2 with the FDR cut-off set to 0.05 (default setting). By this analysis, we obtained 2327 peaks enriched for 18 hpi and 3136 peaks enriched for 4 dpi. For GSEA, all the SOX4 peaks were annotated to the nearest genes using the annotatePeak function of ChIPSeeker R package[57], and the annotated genes were ranked based on the log-fold change between 18 hpi and 4dpi samples calculated by DESeq2, and the ranked gene list was used as input for fgsea R package (fast preranked GSEA)[39].

After confirmation of reproducibility across the three biological replicates by PCA mapping (Supplementary Fig. 11a), we re-performed peak calling using MACS2 with replicate-merged SOX4 samples and the corresponding replicated-merged isotype control without FDR filtering, which generated 9,463 and 19,362 SOX4 peaks at 18 hpi and 4 dpi samples respectively. Motif analysis was performed using HOMER with the option "-size 300 −mask." For visualization with IGV and deepTools, SOX4 BAM files were normalized by subtracting the isotype control signals using the bamCompare function.

Quantification of total SOX4 binding was performed by calculating scale factors for each SOX4 sample using the E. coli spike-in controls. Briefly, the reads were aligned to E. coli genome (K12_MG1655) using Bowtie2, and the scale factors were calculated as the ratios of "(Number of mouse-aligned reads)/(Number of E. coli-aligned reads)".

## CUT&Tag-Seq

CUT&Tag was performed following EpiCypher® CUTANA™ CUT&RUN Protocol v1.7 with minor modification. Normal hepatocytes were harvested by the Percoll density gradient method as described above. Cells at different stages of biliary reprogramming were harvested from the Cas9-EGFP mice challenged with 0.1% DDC for 8-9 weeks by FACS ($n = 2$-3). The cells were washed once in PBS by centrifugation at $600 \times g$ (normal hepatocytes) or $800 \times g$ (reprogrammed/biliary cells) at 4 °C for 3 min. The cells were lightly crosslinked by in 1 ml DMEM containing 0.1–0.15% formalin at RT for 1 min. Crosslinking was quenched by adding 50 µl 2.5 M glycine and rotated at RT for 10 min.

The cells were collected by centrifugation at $600 \times g$ (normal hepatocytes) or $800 \times g$ (reprogrammed/biliary cells) at 4 °C for 3 min, resuspended in 10%FBS-containing DMEM supplemented with 10% DMSO, and frozen down in a Mr. Frosty™ Freezing Container (Thermo) at −80 °C until the day of starting the CUT&Tag experiment.

The cells were thawed in a water bath at 37 °C, collected by centrifugation at $600 \times g$ (normal hepatocytes) or $800 \times g$ (reprogrammed/biliary cells) at RT for 3 min. The supernatant was removed, and the cells were resuspended in nuclear extraction buffer (NE buffer) (20 mM HEPES, pH 7.9, 10 mM KCl, 0.1% Triton X-100, 20% glycerol, 0.5 mM spermidine and EDTA-free protease inhibitors) (approximately 100 μl per 100,000 cells), and incubated on ice for 10 min. The extracted nuclei were washed once in NE buffer by centrifugation at $1300 \times g$ at 4 °C for 3 min. The supernatant was removed, and the nuclei were resuspended in NE buffer, and aliquoted into an 8-well strip tube (100 μl per reaction). Then, 10 μl Concanavalin A-coated magnetic beads (EpiCypher) were added, and incubated at RT for 10 min. The nuclei/bead conjugate was bound to a magnet, the supernatant was removed, and were resuspended in 50 μl antibody reaction buffer (Wash150-T buffer with 5% digitonin and 0.5 M EDTA; the composition of Wash150-T buffer is 20 mM HEPES, pH 7.5, 150 mM NaCl, 0.5 mM spermidine, EDTA-free protease inhibitors and 0.05% Triton X-100), 0.5 μl antibody was added (Supplementary Table 3) and incubated at 4 °C overnight. After removal of the supernatant, the beads were resuspended in 50 μl Wash150-T buffer containing 0.5 μl secondary antibody (Supplementary Table 3) and incubated at RT for 30 min. Following two washes in 200 μl cold Wash150-T buffer, beads were resuspended in 50 μl Wash300-T buffer (20 mM HEPES, pH 7.5, 300 mM NaCl, 0.5 mM spermidine, EDTA-free protease inhibitors and 0.05% Triton X-100). Then, 2.5 μl pAG-Tn5 (EpiCypher) was added, and the samples were incubated at RT for 1 h. While on the magnet, the supernatant was removed, and the samples were washed twice in 200 μl cold Wash300-T buffer. Following resuspension in 50 μl Tagmentation buffer (Wash300-T buffer with 10 mM MgCl$_2$ (Sigma)), Tn5 digestion was performed at 37 °C for 1 h. The supernatant was discarded, and the beads were washed once in in 50 μl TAPS buffer (10 mM TAPS (Boston BioProducts), pH 8.5, 0.2 mM EDTA). Tagmentation was quenched by adding 5 μl SDS release buffer (10 mM TAPS, pH 8.5, 0.1% SDS), mixed thoroughly by vortex and incubated at 58 °C for 1 h. Then, 15 μl SDS quench buffer (0.67% Triton X-100) was added and mixed thoroughly by vortex. Then, 2 μl universal i5 primer (10 μM), 2 μl barcode i7 primer (10 μM), and 25 μl CUTANA® High Fidelity 2X PCR Master Mix for CUT&Tag (EpiCypher) was added, and PCR was performed (1 cycle of 20 °C for 5 min; 1 cycle of 72 °C for 5 min; 1 cycle of 98 °C for 45 s, 14–21 cycles of 98 °C for 15 s and 60 °C for 10 s; 72 °C for 1 min). The DNA was cleaned up with 1.3× AMPure XP beads (Beckman) following the standard procedure.

### Bioinformatics for histone post-translational modification CUT&RUN-Seq and CUT&Tag-Seq
CUT&RUN-Seq and CUT&Tag-Seq of histone post-translational modification was performed with either NextSeq500 (75-bp) or NextSeq2000 (65-bp) for CUT&RUN and NextSeq2000 (55-bp) for CUT&Tag. Reads were aligned to the mouse genome (mm10) using Bowtie2 with the option "-X 1000," and duplicates were removed using Picard. For visualization using IGV and deepTools, BAM files were converted to bigwig files using bamCoverage with the "read counts per million (CPM)" normalization method. Once we confirmed the reproducibility across the three biological replicates by PCA mapping (Supplementary Figs. 13d and 15b), we combined the fastq files (for CUT&RUN-Seq, 75-bp data were shortened to 65-bp using Cutadapt) and obtained single bigwig files for each group. Unless otherwise mentioned, all the heatmaps and aggregate plots in this manuscript are shown for the replicate-merged data.

### Immunocytochemistry of HA-SOX4 prior to CUT&RUN-Seq
After the overnight primary antibody reaction as described above, the cells were incubated with 1/300 AlexaFluor 594-conjugated anti-rabbit IgG (Invitrogen) diluted in permeabilization buffer at RT for 1 h. Then, the cells were spread onto a 24 well plate for imaging.

### Immunoprecipitation
*Sox4*-expressing hepatocytes were harvested from Cas9-EGFP mice injected with AAV8-TBG-*HA-Sox4*-P2A-*Cre* ($1 \times 10^{12}$ genome copies/mouse) at 2 dpi. As a negative control, AAV8-EV-injected hepatoctyes were also harvested ($1 \times 10^{12}$ genome copies/mouse, 4 dpi). Approximately $7–10 \times 10^6$ cells were washed once in PBS by centrifuging at $800 \times g$ for 1 min, resuspended in cell lysis buffer (50 mM Tris, pH 7.5, 150 mM NaCl, 0.1%NP-40, 5 mM EDTA and 1× Halt™ Protease and Phosphatase Inhibitor Cocktail (Pierce)), and incubated on ice for 30 min. The samples were then sonicated using the Bioruptor Plus Sonicator (Diagenode) with Medium Power for 5 cycles (30 s ON and 30 s OFF for 5 min per cycle). After centrifuging at $14,000 \times g$ at 4 °C for 15 min, the supernatant was transferred to new tubes. Following measurement of protein concentration using a BCA Protein Assay Kit (Pierce), 500 μg of protein lysates were used for immunoprecipitation, while 3% (15 μg) of the lysates were saved as input. For immunoprecipitation, lysate was incubated with 6.25 μl (62.5 μg) of washed protein A/G magnetic beads (Pierce) for pre-clearing at RT for 15 min with rotation. The beads were bound to magnet, the supernatant was transferred to a new tube containing 12.5 μl (125 μg) of washed anti-HA magnetic beads (Pierce), and then incubated at RT for 30 min with rotation. The protein/beads conjugate were washed 5 times with 300 μl TBS-T (25 mM Tris, pH 7.5, 150 mM NaCl, 0.05% Tween-20), followed by protein elution with 2× Laemmli Sample Buffer (Bio-Rad) with 5% 2-mercaptoethanol (Sigma) for 10 min at 95–100 °C. Samples were then analyzed by western blot.

### SDS-PAGE and western blot analysis
Immunoprecipitated or input lysate samples were separated by SDS-PAGE using 4–20% Mini-PROTEAN® TGX™ Precast Gels (Bio-Rad), and transferred to Immun-Blot® PVDF Membrane (Bio-Rad) at 100 V for 90 min. Membranes were blocked with 5% milk in PBS-T (PBS containing 0.1% Tween 20) at RT for 30 min, and incubated with primary antibodies (anti-HDAC1, 2 or 3) (Supplementary Table 3) diluted in 5% milk in PBS-T at 4 °C overnight. After washing with PBS-T three times, the membranes were incubated with an HRP-conjugated secondary antibody at RT for 1 h. After washing with PBS-T three times, the membrane was incubated with Pierce™ ECL Plus Substrate (Thermo) or SuperSignal™ West Femto Maximum Sensitivity Substrate (Thermo) for 5 min at RT, and imaged with ChemiDoc Imaging Systems (Bio-Rad). Then some of the membranes were incubated with Restore™ Western Blot Stripping Buffer (Thermo) at RT for 15 min. After washing with PBS-T three times, the membranes were blocked with 5% milk in PBS-T at RT for 30 min, and incubated with primary antibodies (anti-HA or anti-GAPDH) at RT for 1 h. After washing with PBS-T three times, the membranes were incubated with the HRP-conjugated secondary antibody at RT for 1 h. After washing with PBS-T three times, the membrane was incubated with Pierce™ ECL Substrate (for detecting HA) or Pierce™ ECL Plus Substrate (for detecting GAPDH) for 5 min at RT, and imaged with ChemiDoc Imaging Systems (Bio-Rad).

### Recombinant HNF4A and SOX4 protein
MYC-FLAG-HNF4A was purchased from OriGene. CMV-FLAG-SOX4 was produced using 10 × 15 cm plates of 293 T cells. Approximately 70% confluent plates were replenished with 15 ml/plate fresh DMEM (Thermo) supplemented with 2% FBS (Thermo) without antibiotics.

For a 15 cm plate, 48 µg CMV-FLAG-SOX4 plasmid and 144 µl of 1 mg/ml PEI were mixed in 9 ml OptiMEM. After incubation at RT for 15 min, plasmid/PEI complex was added to 293 T cells in a dropwise manner, and the plates were gently shaken back and forth to mix the medium evenly. After incubation in a $CO_2$ incubator for 2 days, the cells were harvested by standard trypsinization. After washing twice in 10 ml PBS by centrifugation at $800 \times g$ for 2 min, the cells were resuspended in lysis buffer (50 mM Tris-HCl, pH 7.4, 150 mM NaCl, 1 mM EDTA, 1% TritonX-100, 1× Halt™ Protease and Phosphatase Inhibitor Cocktail (Pierce)) to the final volume of approximately 12 ml. After incubation at RT on a rotator for 30 min, the suspension was split into 300 µl aliquots, and sonicated using the Bioruptor Plus Sonicator (Diagenode) with High Power for 5 cycles (30 s ON and 30 s OFF for 5 min per cycle). Sonicated samples were centrifuged at 4 °C at 14,000 × g for 15 min, and the supernatant was collected and combined into one tube. Small debris were further removed by passing the samples through a 0.45 µm PVDF filter (Millipore). An affinity chromatography column was prepared with -0.6 ml anti-FLAG M2 beads (Sigma) following the manufacturer's instructions. The lysate was loaded onto the column under gravity flow. The column was washed with 12 ml (-20× volume) TBS (50 mM Tris-HCl, 150 mM NaCl, pH 7.4), and FLAG-SOX4 protein was eluted by 5 rounds of competitive elution with 1× column volume (0.6 ml) of 100 µg/ml FLAG peptide (Sigma) in TBS. Eluted FLAG-SOX4 was concentrated to -50 µl using 30 K MWCO columns (Thermo). Finally, the buffer was exchanged with 20 mM Tris-HCl, pH 7.5, 150 mM NaCl, 1 mM DTT using 7 K MWCO Pierce Zeba™ Desalt Spin Columns (Thermo).

For quantification, MYC-FLAG-HNF4A and FLAG-SOX4 were diluted in 2× Laemmli Sample Buffer (Bio-Rad) with 5% 2-mercaptoethanol, denatured for 5 min at 95–100 °C, and separated by SDS-PAGE using a 4–20% Mini-PROTEAN® TGX™ Precast Gel. The concentration of the full-length proteins was estimated by quantifying the band densities corresponding to 50–75 kDa using BSA standards (Pierce) (Supplementary Fig. 14).

## Designing a double stranded DNA oligonucleotide *Cyp2e1* enhancer probe

We selected a motif located in a *Cyp2e1* distal enhancer and made a double stranded (ds-) DNA oligonucleotide with 25-bp length (genomic location: chr7:140,707,906–140,707,930 of GRCm38 (mm10) genome; 16 bp HNF4A motif sequence flanked by 4-bp and 5-bp extension). We also made a mutant ds-DNA oligonucleotide with a scrambled HNF4A motif. These ds-DNA oligonucleotides were synthesized and labeled with Cy5 at their 5′-end (purchased from IDT). The sequences of these ds-DNA oligonucleotides of WT and mutant are 5′-ATCTCTGGGACAAAGTATAGAGAAG and 5′-ATCTCTGAATCATAGAGAGGAGAAG, respectively.

## Nucleosome preparation

The 162 bp *LIN28B* DNA fragment corresponds to the genomic location:
hg18-chr6: 105,638,004–105,638,165
AGTGGTATTAACATATCCTCAGTGGTGAGTATTAACATGGAACTTACT
CCAACAATACAGATGCTGAATAAATGTAGTCTAAGTGAAGGAAGAAG
GAAAGGTGGGAGCTGCCATCACTCAGAATTGTCCAGCAGGGATTGT
GCAA GCTTGTGAATAAAGACA.

The DNA sequence was created by PCR with end-labeled primers: Cy5-*LIN28B*-Fw: 5Cy5/AGTGGTATTAACATATCCTCAGTGGTG; *LIN28B*-Rv: TGTCTTTATTCACAAGCTTGCACAA. The 162 bp fluorescent-tagged DNA fragments were gel extracted. The nucleosomes were reconstituted by salt dilution. Briefly, a reaction mixture was prepared with 10–20 pM labeled DNA fragment and octamer at a 1:1 DNA:histone ratio and diluted to 10 µl in 10 mM Tris-HCl, pH 8.0, 2 M NaCl, and then incubated at RT for 30 min. Then, 3.5, 6.5, 13.5, and 46.5 µl of 10 mM Tris-HCl pH 8.0 was added at 30 min intervals, which brought the

reactions to 1.48, 1.0, 0.6 and 0.25 M NaCl (80 µl total volume). The mononucleosomes were further purified by 10–30% glycerol gradient followed by dialysis with 10 mM Tris-HCl, pH 8.0, and 1 mM 2-mercaptoethanol. The reconstituted nucleosomes were heat-shifted by incubating at 37 °C for 30 min.

## Binding reactions

The end-labeled ds-DNA oligonucleotides (WT or mutant mouse *Cyp2e1* enhancer-DNA, or *LIN28B*-DNA) and *LIN28B*-nucleosomes were incubated with recombinant proteins in DNA-binding buffer (10 mM Tris-HCl (pH7.5), 1 mM $MgCl_2$, 1 mM DTT, 50 mM KCl, 0.3 mg/ml BSA, 5% Glycerol) at RT for 30 min. Free and bound DNA were separated on 5% non-denaturing polyacrylamide gels run in 0.5× Tris−borate−EDTA and visualized using a PhosphorImager using Cy5 fluorescence setting (excitation at 633 nm and emission filter 670 BP 30) and high sensitivity setting.

## Bioinformatics for ChIP-Seq

ChIP-Seq data were downloaded from the GEO database with the accession numbers GSE57559 for H2B and H3[40]; GSE137066 for adult liver HNF4A[42]; GSE53736 for the adult liver RXRA data[61]; GSE29184 for the mouse adult liver H3K4me3[62].

When biological replicates were available, the downloaded fastq files were first combined. Reads were aligned to the mouse genome (mm10) using Bowtie2 with the option "-X 1000," and duplicates were removed using Picard. For visualization using the IGV and deepTools, BAM files were converted to bigwig files using the bamCoverage with the CPM normalization method. Peak calling was performed using MACS2[55] with an FDR of 0.01 (default setting) without control inputs for H2B, H3 and H3K4me3 and with control inputs for liver HNF4A and RXRA.

## Bioinformatics for MNase-Seq

Low- and high-level MNase-Seq data were downloaded from GEO with the accession number GSE57559[40]. Reads were aligned to the mouse genome (mm10) using Bowtie2 with the option "-X 1000," and duplicates were removed using Picard. BAM files of biological replicates were combined using "samtools merge," and inputted into DANPOS3[63] to calculate the nucleosome occupancy. For visualization using the IGV and DeepTools, the DANPOS3-generated BAM files were converted to bigwig files using the bamCoverage with the RPGC normalization method.

## Generation of the list of active liver enhancer loci

Liver-specific active enhancers in this study are defined as genomic regions which are p300+ H3K4me1+ H3K4me3- H3K27ac+ with DNA hypersensitivity (DHS). To obtain the list of these regions, we used a previously published adult mouse liver-specific enhancer list that was identified by Shen and colleagues based on ChIP-Seq data of p300, H3K4me1, H3K4me3 (ENCODE, GSE29184)[62]. Since H3K27ac predominantly marks active enhancers, we filtered these enhancers so that their central 1 kb regions have 300 or more base-pair overlap with adult mouse liver H3K27ac peaks that were identified in the same study (ENCODE, GSE29184)[62]. For filtering DHS-positive adult liver-specific enhancers, we further filtered them so that their central 1 kb regions have 300 or more base-pair overlap with adult mice liver DHS peaks (ENCODE, GSM1014195).

## Generating a list of active liver promoter loci

Active promoters in this study are defined as H3K4me3+ transcription start sites (TSSs) in the adult liver in the proximity of highly expressed genes. H3K4me3+ regions were defined as H3K4me3 ChIP-Seq peaks (GSE29184) with 0.5 kb extension bilaterally. Highly expressed genes in hepatocytes were defined as the genes ranked within the top 25% in normal hepatocytes using the TPM-normalized RNA-Seq data obtained above (GSE218947). The H3K4me3+ regions were annotated with the

nearest genes (output of MACS2), and then filtered with the list of hepatocyte-highly expressed genes. TSSs included in these regions were regarded as active liver promoter loci ($n = 13,458$).

## Manuscript preparation

Graphics in Figs. 1–3, 5, and 8 were created with Biorender.com and reproduced here with permission.

## Reporting summary

Further information on research design is available in the Nature Portfolio Reporting Summary linked to this article.

## Data availability

Data that generated during the study have been deposited in Gene Expression Ominubus (GEO) with the accession number GSE221225 (SuperSeries) with SubSeries accession numbers GSE218945 (RNA-Seq) and GSE218947 (RNA-Seq); GSE219052 (ATAC-Seq) and GSE253988 (ATAC-Seq); GSE221223 (CUT&RUN-Seq) and GSE221224 (CUT&RUN-Seq). Detailed scripts and parameters used for each step of the analysis will be provided by reasonable request to the authors. Source data are provided with this paper.

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

## Acknowledgements

This work was supported by NIH grants R01DK083355 (B.Z.S.), R01DK125387 (K.S.Z.), R01CA249210 (I.A.A.), Department of Defense grant W81XWH-17-0404 (I.A.A.), the Fred and Suzanne Biesecker Pediatric Liver Center, the Cholangiocarcinoma Foundation, the Abramson Family Cancer Research Institute, the International Medical Research Foundation, the Daiichi Sankyo Foundation of Life Science, the Mochida Memorial Foundation for Medical and Pharmaceutical Research, The Mitsukoshi Health and Welfare Foundation, the Uehara Memorial Foundation, the Kanae Foundation, the Japanese Biochemical Society, and the Osamu Hayaishi Memorial Scholarship for study abroad. We thank the Penn Center for Molecular Studies in Digestive and Liver Disease (P30DK050306) for assistance with tissue processing, Véronique Lefebrvre and Rajan Jain for helpful discussions and comments on the manuscript, Steven Henikoff (Fred Hutchinson Cancer Research Center) for advice on CUT&Tag experiments, Hideharu Hashimoto (Thomas Jefferson University) for advice on purification of SOX4 protein, Tianpeng Zhang (University of Pennsylvania) for advice on immunoprecipitation experiments, and members of the Stanger and Zaret labs for useful suggestions.

## Author contributions

Conceptualization, T.K. and B.Z.S; methodology, T.K., J.S., K.I., A.K., S.Y., J.L., A.J.M., N.T., Q.L., K.S.Z., B.Z.S.; experimental design, T.K., J.S., K.S.Z., B.Z.S.; conduction of experiments, T.K., S.Y., H.C., N.T.; resources, R.U.R, I.A.; bioinformatics analysis, T.K., J.S., K.I., A.K.; writing, T.K., J.S., K.S.Z., B.Z.S.; funding acquisition, K.S.Z., B.Z.S.; supervision, K.S.Z., B.Z.S.

## Competing interests

The authors declare no competing interests.
