## [Peer Review File · Nature Communications]

Cellular reprogramming *in vivo* initiated by SOX4 pioneer factor activityEditorial Note: This manuscript has been previously reviewed at another journal that is not operating a transparent peer review scheme. This document only contains reviewer comments and rebuttal letters for versions considered at *Nature Communications*.

REVIEWER COMMENTS

Reviewer #1 (Remarks to the Author):

I would like to thank the authors for their answer which is extremely detailed and helpful. However, some aspects still need to be reinforced. I also would like to reassure the authors that the “density of the manuscript” has not “contributed to the Reviewer’s confusion”.

The original comments concerning the single cell transcriptomics still stand. The argument that these analyses have already been published and thus reviewed is simply not valid as the fact these analyses only represent a minor part of the manuscript. The authors must explain their QC and other aspects. The mRNA soup is a problem which explains in part the number of cholangiocytes expressing hepatocyte markers. The authors themselves admit that they see many cells expressing transcripts and not protein. This is not a post-transcriptional regulation but a false positive from the single cell analyses. Thus, the process of transdifferentiation is likely less important than suggested in their conclusion.

The authors dismiss the comments on the level of Sox4 not being physiological and inducing cell death. Their argument is not totally rational. They justify that Sox4 is toxic since DCC is also toxic. DCC is a toxin killing liver cells and this results in the induction of Sox5 which in turn initiates the transdifferentiation process. Thus, cells which are surviving this process are the ones able to express Sox4 and not the opposite. Again, they need to admit that Sox4 overexpression is problematic and just include a clear discussion on this point. The fact that they perform their molecular analyses at 4 dpi could be a useful argument.

Furthermore, the limited new data provided to address this comment are not convincing. The staining in 2C does not show co-expression of Sox4 and Sox9. Sox4 staining is not nuclear (they did not provide DAPI) and seems to mark dying cells (again the same problem with toxicity).

The analysis on Sox4 expression is still problematic and somewhat misleading. They use a log scale to hide a major difference in Sox4 expression in most figures describing Sox4 expression. Figure 2g is a good example. This last figure seems to show that Sox4 is 10 times more expressed using AAV overexpression when compared to hepatocytes after DCC induction and in healthy cholangiocytes. This is massive for a transcription factor. They must avoid the use of a log scale and clearly give a fold induction vs control.

The knockout experiments are indeed very useful. However, they seem to suggest that the absence of Sox4 is not really blocking transdifferentiation. Indeed, the difference in the number of CD24-expressing cells is borderline significant while the number of EPCAM-expressing cells is not significant. Sox9 or HNF1b knockouts are much more efficient. Again, they need to discuss this result openly since it strongly suggests a functional hierarchy between these factors.

Concerning the ATAC-seq, the authors must provide the genome track as requested in the original review. This is a very simple request, why being so dismissive? Is there a problem with the quality of the data?

Reviewer #2 (Remarks to the Author):

Katsuda et al considerably rewrote their manuscript and succeeded in making it accessible to the average reader. The addition of substantial new data greatly strengthened the model proposed by the authors, in particular the DNA binding studies and the revised format of the figures. The core of their proposed model is better supported, but a few additional experiments would strengthen the model of sequential repression of hepatocyte genes followed by activation of biliary genes, in particular the former aspect. Finally, the discussion is now minimalist and may be enhanced by discussing a few points as suggested below.

Specific comments

L210-L277. The authors show a loss of H3K27 acetylation at down-regulated putative enhancers following SOX4 action and they provide data for direct interaction between SOX4 and HDAC. Their model of SOX4 directed repression would be strengthened if the authors could perform ChIP-Seq for HDAC1 at different times following SOX4 expression and show co-temporal recruitment of HDAC1 with recruitment of SOX4.

L292-L293. While it is true that some of the authors previously documented recognition of partial binding motifs by pioneer factors, the evidence presented for SOX4 in the present work would rather argue for overlapping strong sites for SOX4 and HNF4 rather than recruitment at weak binding sites. The introduction to the DNA binding studies is thus somewhat misleading despite it being correct.

L294-L302. The apparent stronger binding of SOX4 compared to HNF4A in EMSA is convincing using the recombinant proteins depicted in the work. But these results could be discussed in the Discussion in the perspective that either or both of these proteins may be susceptible to posttranslational modification as a result of signalling pathway action. And indeed, the authors invoke the in vivo role of signalling pathways in the biological reprogramming. It is an interesting hypothesis that signal-dependent modification of either SOX4 and/or HNF4A could alter their DNA binding ability and/or their ability to recruit coactivators or corepressors. Discussing these possibilities would, in this reviewer's opinion, enhance the Discussion.

In fact, such discussion would be far more substantial than the relatively bland statement made on L377 and L378 of the Discussion and give real substance to the present comment.

In summary, this revised manuscript is greatly improved and is now suitable for publication pending minor revision.

Reviewer #3 (Remarks to the Author):

The authors did an excellent job in revising the manuscript. All points raised by this reviewer were satisfactorily addressed.

I would recommend the manuscript for publication.

Reviewer #1 (Remarks to the Author):

I would like to thank the authors for their answer which is extremely detailed and helpful. However, some aspects still need to be reinforced. I also would like to reassure the authors that the “density of the manuscript” has not “contributed to the Reviewer’s confusion”.

Response: We thank the Reviewer for reviewing our paper again. It is good for us to know that our detailed answers were helpful. We have addressed all the newly raised comments as described below.

The original comments concerning the single cell transcriptomics still stand. The argument that these analyses have already been published and thus reviewed is simply not valid as the fact these analyses only represent a minor part of the manuscript. The authors must explain their QC and other aspects. The mRNA soup is a problem which explains in part the number of cholangiocytes expressing hepatocyte markers. The authors themselves admit that they see many cells expressing transcripts and not protein. This is not a post-transcriptional regulation but a false positive from the single cell analyses. Thus, the process of transdifferentiation is likely less important than suggested in their conclusion.

Response: We consider that this comment is no longer relevant, since our revised manuscript no longer has any scRNA-seq data, nor do we reference or discuss previously published scRNA-seq data. Given that there is no mention of single-cell data or analysis, this comment has been fully addressed without additional changes.

The authors dismiss the comments on the level of Sox4 not being physiological and inducing cell death. Their argument is not totally rational. They justify that Sox4 is toxic since DDC is also toxic. DDC is a toxin killing liver cells and this results in the induction of Sox5 which in turn initiates the transdifferentiation process. Thus, cells which are surviving this process are the ones able to express Sox4 and not the opposite. Again, they need to admit that Sox4 overexpression is problematic and just include a clear discussion on this point. The fact that they perform their molecular analyses at 4 dpi could be a useful argument.

Response: We appreciate the Reviewer’s continued concern over the relevance of Sox4 in our experimental set up. We admit that exogenous expression of Sox4 is higher than in the endogenous setting (DDC), and that it is technically challenging to quantify the extent of similarity between the toxicity caused by exogenous Sox4 versus DDC. However, correlation analyses using the RNA-Seq and ATAC-Seq data strongly indicate that Sox4-expressing hepatocytes are closely clustered with Rep_early cells in the DDC setting (we added new data in revised **Supplementary Fig. 8b, Supplementary Fig. 9c**). Indeed, on the gene expression level, the Pearson’s correlation coefficient between the Sox4 condition and the Rep_early condition are comparable to the intragroup correlation between either condition. This indicates a very high degree of similarity between these conditions. An additional point is that the fold induction of Sox4 during DDC reprogramming is greater than that for Sox9, and similarly our AAV delivery of Sox4 is greater than that of Sox9. Taking everything together, and the reviewer’s point about focusing on 4dpi, we maintain our conclusion that Sox4-expressing cells substantially recapitulate the *early* stage of reprogramming.

Furthermore, the limited new data provided to address this comment are not convincing. The staining in 2C does not show co-expression of Sox4 and Sox9. Sox 4 staining is not nuclear (they did not provide DAPI) and seems to mark dying cells (again the same problem with toxicity).

Response: We have now included DAPI staining images in **revised Fig. 2c**. Given that the background signals of SOX4 were high, which could compromise the visibility of nuclei staining, we have also included images of each channel individually in **revised Supplementary Fig. 5a**. In addition, we also include HA IF images of AAV-HA-Sox4 liver, which more clearly shows nuclear expression of SOX4 protein (revised **Supplementary Fig. 5b**).

The analysis on Sox4 expression is still problematic and somewhat misleading. They use log scale to hide major difference in Sox4 expression in most figure describing Sox4 expression. Figure 2g is a good example. This last figure seems to show that Sox4 is 10 time more expressed using AAV overexpression when compared hepatocytes after DCC induction and in healthy cholangiocytes. This is massive for a transcription factor. They must avoid to the use of log scale and clearly give a fold induction vs control.

Response: The data is presented in manuscript as clearly and transparently as possible, while avoiding redundancy which would further increase the density of the manuscript. While we maintain that the log scale is useful for visualization, we appreciate the Reviewer’s point that there is value in this case of using the non-log-scale. Therefore, we added a panel with its y-axis linear-scaled in **revised Supplementary Figure 3a** to complement the panel with its y-axis log-scaled. As this panel indicates, the fold change between Rep_intermed or Rep_late compared to AAV-HA-Sox4 cells is less than 2.5 times. We admit that the expression level is higher in the AAV-injected setting, but we consider this to represent a comparable level rather than “over-expression” of “10 times” as the Reviewer suggests. Moreover, when we tested whether the normalized expression of Sox4 in the Hep_Sox4 condition was greater than each of the other conditions, using a one-sided t-test with Benjamini-Hochburg correction for multiple hypothesis testing, we did not find statistically significant differences between Hep_Sox4 and any of Rep_intermed, Rep_late and BEC groups, while we confirmed that the expression is higher than the EV and Rep_Early conditions (**the graph below**, which does not appear in the manuscript). Concordant with our findings, it should be also noted that Anshul Kundaje has a preprint on bioRxiv doing single cell analysis of reprogramming to iPS and shows that only the cells expressing the very highest levels of OSKM go on to reprogram (doi: <https://doi.org/10.1101/2023.10.04.560808>). And once again we note that the fold induction of Sox4 during DDC reprogramming is greater than that for Sox9, and similarly our AAV delivery of Sox4 is greater than that of Sox9.

Figure for the rebuttal.
 RNA-Seq data of Sox4. The expression levels are compared with the Hep_Sox4 group as the reference. * p_adj < 0.05; ns, not significant

The knockout experiments are indeed very useful. However, they seem to suggest that absence of Sox4 is not really blocking transdifferentiation. Indeed, the difference in number of CD24 expressing cells is borderline significant while the number of EPCAM expressing cells is not significant. Sox9 or HNF1b knock out are much more efficient. Again, they need to discuss this result openly since it strongly suggests a functional hierarchy between these factors.

Response: We have added a few sentences to discuss this issue by focusing on apparent redundant mechanisms for the SOX4 pioneer activity as follows (**lines 414-421**):

“Moreover, given that necessity for Sox4 was weak in the loss-of-function study, pioneer activity of Sox4 may be compensated for by other transcription factors. We did not observe compensatory upregulation of Sox11 or Sox12 (other SoxC family members) in Sox4-KO hepatocytes by RNA-Seq (data not shown). Therefore, certain transcription factors other than Sox family members could have compensated for *the* Sox4 loss. Thus, further studies are expected to elucidate the transcriptional regulatory mechanisms underlying the transitions to the later stages of the metaplastic process, as well as the compensatory mechanism for pioneer activity of Sox4.”

Concerning the ATAC-Seq, they author must provide the genome track as request in the original review. This is very simpler request why being so dismissive? Is there a problem with the quality of the data?

Response: As per the reviewer’s request, we have provided ATAC-Seq tracks in **revised Supplementary Fig. 10**.

Reviewer #2 (Remarks to the Author):

Katsuda et al considerably rewrote their manuscript and succeeded in making it accessible to the average reader. The addition of substantial new data greatly strengthened the model proposed by the authors, in particular the DNA binding studies and the revised format of the figures. The core of their proposed model is better supported, but a few additional experiments would strengthen the model of sequential repression of hepatocyte genes followed by activation of biliary genes, in particular the former aspect. Finally, the discussion is now minimalist and may be enhanced by discussing a few points as suggested below.

Response: We thank the Reviewer for the very positive comments and constructive suggestions. We have addressed all the newly raised issues by additional discussion.

Specific comments

L210-L277. The authors show a loss of H3K27 acetylation at down-regulated putative enhancers following SOX4 action and they provide data for direct interaction between SOX4 and HDAC. Their model of SOX4 directed repression would be strengthened if the authors could perform ChIP-Seq for HDAC1 at different times following SOX4 expression and show co-temporal recruitment of HDAC1 with recruitment of SOX4.

Response: We thank the Reviewer for this constructive comment. Given the suggested experiment would require a substantial amount of labor and time, we believe that this experiment is beyond the scope of the present study. Nonetheless, since we acknowledge the importance of exploration for the HDAC1 binding sites, we have added a few sentences in Discussion as follows (**lines 381-383**):

“Our study also confirmed the direct association of SOX4 and HDAC1 via their protein-protein interactions. We expect future studies will further investigate how the distribution of HDAC1 and other potential HDACs are altered by SOX4.”

L292-L293. While it is true that some of the authors previously documented recognition of partial binding motifs by pioneer factors, the evidence presented for SOX4 in the present work would rather argue for overlapping strong sites for SOX4 and HNF4 rather than recruitment at weak binding sites. The introduction to the DNA binding studies is thus somewhat misleading despite it being correct.

Response: We agree with this Reviewer’s suggestion to improve the introduction to the relevant section. Accordingly, we changed the description “Considering that pioneer factors have been demonstrated to bind partial motifs, we hypothesized...” to a simpler description “These observations prompted us to hypothesize...” (**lines 296-297**).

L294-L302. The apparent stronger binding of SOX4 compared to HNF4A in EMSA is convincing using the recombinant proteins depicted in the work. But these results could be discussed in the Discussion in the perspective that either or both of these proteins may be susceptible to posttranslational modification as a result of signalling pathway action. And indeed, the authors invoke the in vivo role of signalling pathways in the biological reprogramming. It is an interesting hypothesis that signal-dependent modification of either SOX4 and/or HNF4A could alter their DNA binding ability and/or their ability to recruit coactivators or corepressors. Discussing these possibilities would, in this reviewer’s opinion, enhance the Discussion. In fact, such discussion would be far more substantial than the relatively bland statement made on L377 and L378 of the Discussion and give real substance to the present comment.

Response: We thank the reviewer again for this constructive suggestion. We expanded our consideration of this possibility in the Discussion as follows (**lines 399-402**): “Meanwhile, the mechanism by which SOX4 adopts the role of a transcription activator or repressor is unclear. One possible hypothesis is that post-transcriptional modification of SOX4 could change its role, which we expect to be tested in future studies.”

In summary, this revised manuscript is greatly improved and is now suitable for publication pending minor revision.

Response: We appreciate the reviewer’s positive response and constructive comments.

Reviewer #3 (Remarks to the Author):

The authors did an excellent job in revising the manuscript. All points raised by this reviewer were satisfactorily addressed.

I would recommend the manuscript for publication.

Response: We appreciate the positive response.

REVIEWERS' COMMENTS

Reviewer #1 (Remarks to the Author):

The authors have answered most of my comments. The revised Supplementary Figure 3a does not seem to show expression of Sox4 using linear scale? Finally, the staining on Figure 2C and revised Supplementary Fig. 5a is still problematic. It could be deleted from the manuscript.

Response to Reviewers

Reviewer #1 (Remarks to the Author):

The authors have answered most of my comments. The revised Supplementary Figure 3a does not seem to show expression of Sox4 using linear scale? Finally, the staining on Figure 2C and revised Supplementary Fig. 5a is still problematic. It could be deleted from the manuscript.

Response: The Reviewer correctly points out that the linear scale depiction of Sox4 expression is not contained in Supp. Fig. 3a but rather in Supp. Fig. 4a. We have corrected this typographical error in the manuscript. We have not changed Figs. 2C and Supp. Fig. 5a given editorial approval.